# Degradation of engulfed mitochondria is rate-limiting in Optineurin-mediated mitophagy in neurons

Chantell S Evans, Erika LF Holzbaur*

Department of Physiology, Perelman School of Medicine, University of Pennsylvania, Philadelphia, United States

**Abstract** Mitophagy, the selective removal of damaged mitochondria, is thought to be critical to maintain neuronal homeostasis. Mutations of proteins in the pathway cause neurodegenerative diseases, suggesting defective mitochondrial turnover contributes to neurodegeneration. In primary rat hippocampal neurons, we developed a mitophagy induction paradigm where mild oxidative stress induced low levels of mitochondrial damage. Mitophagy-associated proteins were sequentially recruited to depolarized mitochondria followed by sequestration into autophagosomes. The localization of these mitophagy events had a robust somal bias. In basal and induced conditions, engulfed mitochondria remained in non-acidified organelles for hours to days, illustrating efficient autophagosome sequestration but delayed lysosomal fusion or acidification. Furthermore, expression of an ALS-linked mutation in the pathway disrupted mitochondrial network integrity and this effect was exacerbated by oxidative stress. Thus, age-related decline in neuronal health or expression of disease-associated mutations in the pathway may exacerbate the slow kinetics of neuronal mitophagy, leading to neurodegeneration.

*For correspondence:
holzbaur@pennmedicine.upenn.edu

**Competing interests:** The authors declare that no competing interests exist.

## Introduction

The mitochondrial network is a dynamic interconnected system of organelles undergoing continuous renewal and rearrangement to support cellular needs. Given that neurons have high-energy demands at sites far from the soma, the targeted delivery and regulated removal of mitochondria are essential at these distal sites (*Millecamps and Julien, 2013*; *Sheng, 2014*). These organelles utilize mitochondrial membrane potential to perform cellular functions, including energy generation and $Ca^{2+}$ buffering. Mitochondrial damage or depolarization decreases function and signals maintenance mechanisms to repair or eliminate damaged organelles.

Due to the unique characteristics of neurons, multiple quality control mechanisms likely function independently to maintain mitochondrial health (*Evans and Holzbaur, 2019*). In one pathway, mitochondrial fragments are engulfed in autophagosomes via basal, non-selective autophagy that initiates in the distal axon (*Maday and Holzbaur, 2014*; *Wong and Holzbaur, 2014b*). Autophagy, a conserved lysosomal degradation pathway, involves the engulfment of cargo to be degraded by a double-membrane organelle. Autophagosomes mature during trafficking along the axon to the soma by fusion with lysosomes, which leads to efficient cargo degradation (*Cheng et al., 2015*; *Hollenbeck, 1993*; *Lee et al., 2011*; *Maday and Holzbaur, 2014*; *Maday et al., 2012*; *Neisch et al., 2017*). In a second maintenance mechanism, dysfunctional mitochondria are removed through selective mitophagy, where damaged organelles are specifically sequestered and eliminated from the cell, also via the autophagic machinery (*Harper et al., 2018*; *Misgeld and Schwarz, 2017*; *Nguyen et al., 2016*; *Wang and Klionsky, 2011*; *Youle and Narendra, 2011*). This pathway has been implicated in several neurodegenerative diseases including Parkinson's disease (PD) and amyotrophic lateral sclerosis (ALS; *Evans and Holzbaur, 2019*).

Mitophagy is primarily regulated by two proteins linked to familial PD, PTEN-Induced Putative Kinase 1 (PINK1) and Parkin (*Kitada et al., 1998*; *Narendra et al., 2008*; *Narendra et al., 2010b*; *Valente et al., 2004*; *Wang and Klionsky, 2011*; *Youle and Narendra, 2011*). PINK1 recruits Parkin to phospho-ubiquitinate outer mitochondrial membrane proteins, which act as a specific cue for organelle degradation (*Geisler et al., 2010*; *Kane et al., 2014*; *Kazlauskaite et al., 2014*; *Koyano et al., 2014*; *Matsuda et al., 2010*; *Narendra et al., 2008*; *Narendra et al., 2010b*; *Ordureau et al., 2015*; *Ordureau et al., 2014*; *Poole et al., 2010*; *Sarraf et al., 2013*; *Vives-Bauza et al., 2010*). Optineurin (OPTN), an autophagy receptor causative for glaucoma and ALS (*Maruyama and Kawakami, 2013*; *Maruyama et al., 2010*; *Rezaie et al., 2002*), is recruited to the mitochondrial surface where it interacts with ubiquitin through its ubiquitin binding in ABIN and NEMO (UBAN) domain (*Gleason et al., 2011*; *Heo et al., 2015*; *Lazarou et al., 2015*; *Moore and Holzbaur, 2016*; *Rogov et al., 2014*; *Wild et al., 2011*; *Wong and Holzbaur, 2014a*; *Zhu et al., 2007*). Other mitophagy receptors, such as Nuclear Dot Protein 52 kDa (NDP52) and TAX1 Binding Protein 1 (TAX1BP1), may be recruited in parallel, downstream from Parkin activity (*Heo et al., 2015*; *Lazarou et al., 2015*; *Moore and Holzbaur, 2016*). OPTN association with the damaged mitochondrial surface is stabilized though phosphorylation by TANK-Binding Kinase 1 (TBK1), a serine/threonine kinase mutated in familial ALS (*Freischmidt et al., 2015*; *Heo et al., 2015*; *Richter et al., 2016*; *Wild et al., 2011*). Subsequently, microtubule-associated protein light chain 3 (LC3), is targeted to the ubiquitinated mitochondria via the LC3 interacting region (LIR) of OPTN (*Stolz et al., 2014*; *Wild et al., 2011*), leading to engulfment of the organelle inside an LC3-positive autophagosome. Thus, OPTN is specifically recruited to depolarized organelles to mediate sequestration by an autophagosome and eventual degradation following lysosomal fusion.

Initial work to characterize mitophagy was performed in non-polarized cell types, such as HeLa cells, with subsequent studies focused on analyzing this mechanism in neurons. These results have been controversial, and the relative contribution of PINK1/Parkin-dependent mitophagy to neuronal health in vivo remains unclear. The mitophagy pathway in primary neuronal cultures appears to utilize the same molecular players described in non-neuronal cell lines, suggesting that PINK1/Parkin-mediated mitophagy is a conserved process across cell types. Surprisingly, however, in vivo studies have not fully supported this idea. PINK1 and Parkin knock-out (KO) mice display mild phenotypes with no dopaminergic neuron loss, a hallmark of PD, implying there are additional compensatory pathways contributing to neuronal mitophagy in vivo (*Akundi et al., 2011*; *Goldberg et al., 2003*; *Kitada et al., 2007*; *Perez and Palmiter, 2005*). In sharp contrast, KO rat models of PINK1 exhibited significant dopaminergic neuron loss, and null or mutant flies of PINK1 or Parkin display morphological defects in dopaminergic neurons (*Cha et al., 2005*; *Clark et al., 2006*; *Dave et al., 2014*; *Park et al., 2006*; *Whitworth et al., 2005*; *Yang et al., 2006*). The use of the acid-sensitive *mito*-QC mitophagy probe also did not reveal deficits in tissues of high metabolic demand in mice (*mito*-QC$^{+/+}$ *Pink1*$^{-/-}$) or in dopaminergic neurons of PINK1 and Parkin mutant flies (*Lee et al., 2018*; *McWilliams et al., 2018*). However, the use of another acid-sensitive mt-Keima probe illustrated insufficiencies in basal mitophagy in dopaminergic neurons of Parkin-deficient flies (*Cornelissen et al., 2018*; *Katayama et al., 2011*; *Sun et al., 2015*). Using the same probe in non-neuronal cells, essential roles for PINK1 and Parkin were shown in mitophagy following exhaustive exercise, hypoxic exposure, or rotenone treatment (*Kim et al., 2019*; *Sliter et al., 2018*). Further, stable isotope labeling of flies showed a slowing in the turnover of mitochondrial proteins isolated from the heads of Parkin null flies (*Vincow et al., 2013*). Interestingly, recent work described an inflammatory phenotype in PINK1 and Parkin KO mice after acute or chronic mitochondrial stress that could be rescued by depletion of a regulator of the type 1 IFN signaling response (*Sliter et al., 2018*). Thus, the respective roles of PINK1 and Parkin in mitophagy versus inflammation remain unclear, making it important to more fully characterize this pathway in neurons.

Neurons are highly polarized cells with extended axonal and dendritic processes and significant local energy demands (*Misgeld and Schwarz, 2017*). This cellular morphology results in unique challenges for the cell and raises interesting questions as to whether all mitochondria are subject to the same quality control mechanisms and whether neuronal mitophagy is spatially regulated. In primary cortical neurons, the use of cyanide m-chlorophenylhydrazone (CCCP) treatment to induce mitochondrial damage resulted in an accumulation of Parkin-positive mitochondria in somatodendritic compartments, where damaged organelles underwent autophagosome engulfment and lysosomal degradation (*Cai et al., 2012*). Neurons treated with CCCP or Antimycin A (AA; a potent complex III

inhibitor) displayed enhanced retrograde mitochondrial motility, suggesting damaged organelles are trafficked back to the soma for degradation (*Cai et al., 2012*; *Miller and Sheetz, 2004*). This finding is supported by a recent report in which photo-converted axonal mitochondria underwent mitophagy in the soma after oxygen/glucose deprivation and reperfusion (*Zheng et al., 2019*). In contrast, a study using hippocampal neurons characterized axonal mitophagy induced by treatment with AA or by selective local damage using the construct mito-KillerRed, a genetically encoded photosensitizer that causes light-induced reactive oxygen species (ROS) generation (*Ashrafi et al., 2014*). Clearance of axonal mitochondria was reported to be locally mediated by the autophagic pathway and dependent on PINK1 and Parkin (*Ashrafi et al., 2014*). In addition, several reports have described arrested axonal mitochondrial motility after damage due to the removal of Miro, a mitochondrial motor adaptor protein (*Hsieh et al., 2016*; *Liu et al., 2012*; *Wang et al., 2011*). These later findings have led to a model in which damaged mitochondria remain stationary to allow for efficient local degradation and to prevent dissemination of damaged components through fusion, but consensus on this point is lacking (*Zheng et al., 2019*). However, in vivo studies of mitophagy in PINK1 or Parkin mutant flies found no enhanced accumulation of axonal mitochondria in motor neurons, but did report dramatic changes in the morphology of somal mitochondria, indicating neuronal mitophagy may be compartmentally restricted (*Devireddy et al., 2015*; *Sung et al., 2016*). Furthermore, analysis of basal mitophagy in Purkinje cells from *mito*-QC mice illustrated mitophagic events in somatodendritic compartments, with no axonal events reported (*McWilliams et al., 2016*).

One potential reason for the apparent disconnect between in vitro and in vivo studies, as well as the lack of consensus among studies using primary neurons may be the methods used to initiate mitochondrial damage, including treatments with reagents that are highly toxic to neuronal health. Here, we developed a protocol in which mild oxidative stress was used to induce low levels of mitochondrial damage in primary hippocampal neurons without compromising the entire neuronal network, allowing us to readily visualize turnover under conditions in which stressed cells remain viable. Live-cell imaging was used to visualize the spatiotemporal dynamics of mitophagy-associated protein recruitment to damaged mitochondrial fragments. We found a pronounced accumulation of OPTN-positive mitochondria in the somal compartment of stressed neurons. In contrast, OPTN recruitment was rarely visualized in either dendrites or axons. The time course of mitophagy appears to be much slower in neurons than other cell types, with acidification and clearance occurring many hours after initial damage. Expression of an ALS-associated mutation in OPTN is sufficient to disrupt mitochondrial health under basal conditions and this is exacerbated by oxidative stress. Together, these data indicate that OPTN-dependent mitophagy is an important pathway for both the homeostatic regulation of the somal mitochondrial network and the cellular response to stress.

## Results

### Low-level induction of ROS leads to selective mitochondrial damage without compromising the overall mitochondrial network

We used multicolor live-cell imaging to examine the spatial and temporal dynamics of mitophagy following mitochondrial damage. To model the subtle increases in ROS experienced by highly metabolically active neurons in vivo, we induced mitochondrial damage in hippocampal neurons by mild oxidative stress via antioxidant (AO) deprivation for 1 or 6 hr, noted as AO-free (*Joselin et al., 2012*). In parallel, a low dose of AA (3 nM) was used as a positive control (*Figure 1A*). Cell viability limited the time frame of AA treatment, but since AA is a more potent damaging agent, the 2 h AA time point was compared to the 6 h time points from either AO-free or control conditions. None of these conditions caused widespread cell death (>90% of transfected cells remained viable) and treated neurons recovered within 24 h after treatment.

To quantitatively assess whether these mild oxidative stresses initiated mitochondrial damage, intracellular ROS was detected by a fluorogenic probe, CellROX. The signal intensity of CellROX was low in control neurons and increased significantly in both the AO-free and AA conditions (*Figure 1B*). Since rounding and fragmentation are hallmarks of mitochondrial damage (*Twig et al., 2008*; *Westermann, 2010*; *Youle and Narendra, 2011*), we examined morphology following treatments by either staining with tetramethyl rhodamine ethyl ester (TMRE), a vital dye used to measure mitochondrial potential (*Figure 1C–D*), or by transient transfection with Mito-DsRed (*Figure 1E–F*).

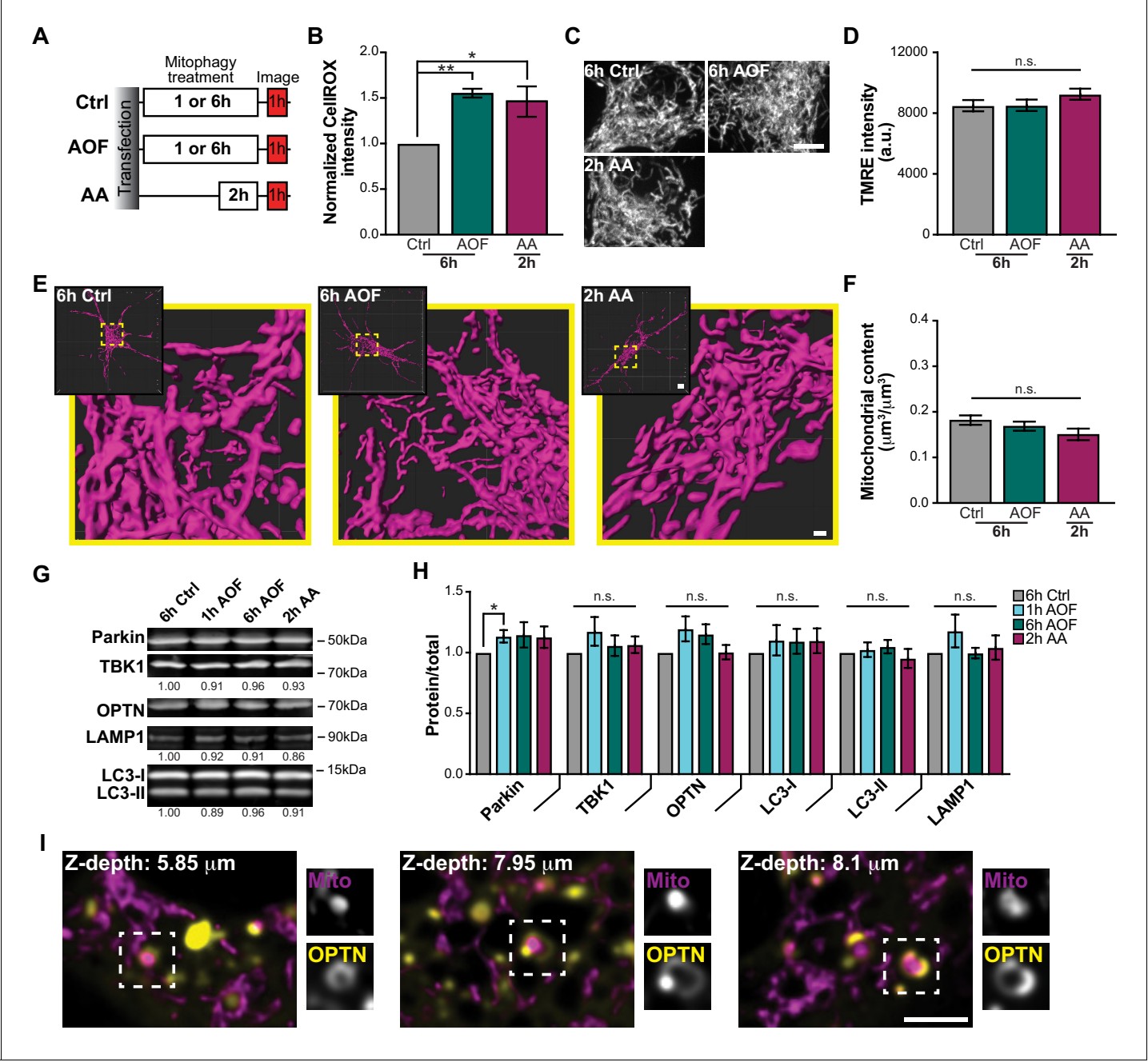

**Figure 1.** Antioxidant removal induces low levels of mitochondrial damage and sequestration without compromising the entire neuronal network. (A) Schematic of experimental paradigm to initiate mitophagy. (B) Fluorescence intensity quantification of intracellular ROS by the CellROX reagent in control and treated conditions. Mean ± SEM; $n$ = 8 wells/condition per replicate, from 4 biological replicates; 7 DIV. *, $p < 0.05$; **, $p < 0.01$ by Kruskal-Wallis ANOVA with Dunn's multiple comparisons test. (C–D) Representative images (C) and quantification (D) of TMRE fluorescence intensity. Mean ± SEM; $n$ = 31-38 neurons from 3-4 biological replicates; 7 DIV. Not significant (n.s.) by Kruskal-Wallis ANOVA with Dunn's multiple comparisons test. Scale bar, 5 μm. (E) Volume renderings of the somal mitochondrial network; original and enlarged images are shown for each neuron. Scale bar, 0.7 μm; inset, 4 μm. (F) Quantification of the somal mitochondrial content. Mean ± SEM; $n$ = 16-24 neurons from 3-4 biological replicates; 6-7 DIV. Not significant (n.s.) by Kruskal-Wallis ANOVA with Dunn's multiple comparisons test. (G–H) Representative Western blot (G) and quantification (H) of mitophagy-associated proteins from cultured hippocampal neurons. Data shown as the fold change over control of the protein of interest divided by total protein stain. Normalization factors are shown under representative images. Mean ± SEM; $n$ = 5 biological replicates; 7-8 DIV. Not significant (n.s.); *, $p < 0.05$ by Kruskal-Wallis ANOVA with Dunn's multiple comparisons test. (I) Representative single plane images from a somal z-stack showing OPTN sequestration of damaged spherical mitochondria; examples of these mitophagy events are shown in insets. Scale bar, 3 μm.

The online version of this article includes the following figure supplement(s) for figure 1:

*Figure 1 continued on next page*

*Figure 1 continued*

**Figure supplement 1.** OPTN puncta localize to rounded and fragmented mitochondria.

Control and treated neurons all displayed dynamic interconnected mitochondrial networks (*Figure 1C and E*). As a more sensitive measure, we quantified the mitochondrial aspect ratio from single z-planes and saw a modest increase in mitochondria with an aspect ratio of ≤1.5 in neurons treated with AO-free media compared to control (*Figure 1—figure supplement 1A*). Quantification of the TMRE intensity or somal mitochondrial content revealed similar levels across the different treatments (*Figure 1D and F*). Thus, large-scale changes in mitochondrial morphology or polarization state were not detected in the somal mitochondrial network following stress.

In neurons, we examined the endogenous expression levels of mitophagy-associated proteins, including Parkin, TBK1, OPTN, LC3, and LAMP1 (*Heo et al., 2015*; *Lazarou et al., 2015*; *Moore and Holzbaur, 2016*; *Narendra et al., 2008*; *Richter et al., 2016*; *Wong and Holzbaur, 2014a*). Protein levels remained consistent across treatments, with the exception of an increase in Parkin expression following 1 h AO-free treatment (*Figure 1G–H*), although it should be noted that the magnitude of the increase was comparable to the nonsignificant increases in Parkin levels following other mitophagy treatments. We also observed no significant differences in the expression levels of ATG5 and ATG16L1, proteins required for autophagy initiation (*Figure 1—figure supplement 1B-C*; *Mizushima et al., 2001*; *Suzuki et al., 2001*). The lack of change in endogenous protein expression suggests that under these conditions, mitophagy is driven by the translocation of existing proteins to damaged mitochondria rather than de novo protein synthesis, which correlates with the time-sensitive nature of this mechanism. However, it is possible that small changes in protein levels due to mitophagic flux cannot be detected via Western blot analysis.

We next determined whether our mitochondrial damage paradigm was sufficient to not only induce organelle damage, but to initiate neuronal mitophagy. In HeLa cells expressing exogenous Parkin, OPTN was efficiently recruited to fragmented mitochondria within 30 min following CCCP treatment (*Moore and Holzbaur, 2016*; *Wong and Holzbaur, 2014a*). In contrast, Parkin is endogenously expressed in hippocampal neurons (*Figure 1G–H*), so Parkin overexpression was not utilized in these experiments unless explicitly stated, to ensure that overexpression did not hyperactivate the pathway leading to non-physiological levels of mitophagy. Cells were transfected with Mito-DsRed and Halo-OPTN and treated for 1 h with control or AO-free media. Monitoring across focal planes using z-stack images, we visualized OPTN rings formed around small spherical mitochondrial fragments distributed throughout the soma (*Figure 1I* and *Video 1*). Z-stack images of the entire soma ensured that OPTN-positive mitochondria were indeed spherical and isolated from the network.

To investigate the extent of OPTN association with damaged mitochondria, we quantified the percent of OPTN puncta on linear (i.e. healthy) versus rounded (i.e. damaged) mitochondria. OPTN association with rounded mitochondria significantly increased after mitophagy induction with AO-free treatment compared to control. In contrast, the percentage of OPTN puncta on linear mitochondria was similar in either condition (*Figure 1—figure supplement 1D*). It should be noted that the majority of OPTN puncta (~60–70%) were not associated with mitochondria. These puncta may represent either the self-association of OPTN or the recruitment of OPTN to other autophagy substrates, such as protein aggregates (*Korac et al., 2013*). Of note, OPTN puncta were also observed in cells expressing endogenous levels of OPTN by immunocytochemistry, indicating they are not a consequence of overexpression.

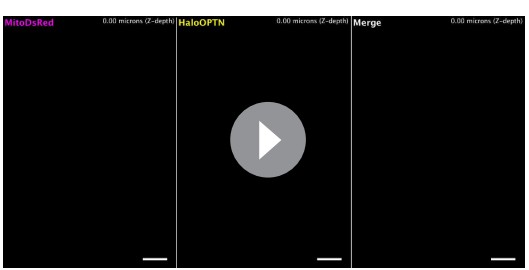

**Video 1.** OPTN is recruited to spherical, damaged mitochondria in the soma. Representative neuron treated for 1 h with AO-free media. Live-cell imaging was used to take a z-stack through the soma. Arrows indicate OPTN rings around damaged spherical mitochondria (see *Figure 1I* for single plane images). Scale bar, 5 μm.
https://elifesciences.org/articles/50260#video1

## OPTN and TBK1 are specifically recruited to damaged mitochondria in hippocampal neurons

In non-neuronal cells, OPTN, NDP52, and TAX1BP1 are proposed to mediate the autophagic clearance of damaged mitochondria, while p62 clusters organelles without directly facilitating mitophagy (*Heo et al., 2015*; *Lazarou et al., 2015*; *Moore and Holzbaur, 2016*; *Narendra et al., 2010a*; *Wong and Holzbaur, 2014a*). In neurons, the extent to which these autophagy receptors cooperate to mediate neuronal mitophagy is unclear. We compared the endogenous protein levels of OPTN, NDP52, TAX1BP1 and p62. Western blot analysis showed that all autophagy receptors were expressed in total brain, cortex and hippocampus (*Figure 2—figure supplement 1A–B*). Immunostaining of endogenous OPTN and p62 indicated that both proteins were found in all neuronal compartments (*Figure 2—figure supplement 1C–H*). The high expression levels of OPTN in human brain samples imply it is the predominate autophagy receptor mediating neuronal mitophagy (*Lazarou et al., 2015*).

The current model suggests that following damage to a mitochondrion, phospho-ubiquitination of outer mitochondrial membrane proteins mediated by PINK1-recruited Parkin increases the abundance of ubiquitin chains, which recruit OPTN and TBK1 to the organelle (*Figure 2A*; *Harper et al., 2018*; *Ordureau et al., 2018*; *Richter et al., 2016*). We confirmed that the observed translocation of OPTN to damaged mitochondria in hippocampal neurons was downstream of Parkin by depleting the E3-ubiquitin ligase using siRNA. In a Parkin knock-down (KD) background, overexpression of mCherry-Parkin$^{WT}$ significantly increased the percentage of cells with OPTN-positive mitochondria rings following 1 h AO-free treatment compared to KD alone (*Figure 2—figure supplement 2*). Only partial rescue was observed with the Parkinson's disease-linked mutant mCherry-Parkin$^{T240R}$ (*Sriram et al., 2005*).

Since OPTN recruitment to mitochondrial fragments was seen following mitophagy treatment (*Figure 1I* and *Figure 2B*), we asked whether OPTN translocation is specific to dysfunctional organelles. After mitophagy induction by mild oxidative stress, most mitochondria exhibited normal dynamics and were dual-labeled with Mito-DsRed and TMRE. In contrast, OPTN-positive mitochondria were labeled with Mito-DsRed but displayed little to no TMRE signal, demonstrating specific OPTN recruitment to damaged organelles (*Figure 2C*). Additionally, we noted that the mitophagy-associated proteins Parkin and TBK1 robustly colocalized along with OPTN to damaged, ubiquitinated organelles (*Figure 2D–F*). These proteins formed rings around damaged mitochondria as observed in confocal slices (*Figure 2D–F*; red arrows and insets, 1); z-stacks highlight the spherical decoration of mitochondrial fragments with these proteins. We also observed puncta of OPTN and other proteins that colocalized with isolated mitochondrial fragments (*Figure 2D–F*; green arrows and insets, 2); these puncta may represent the initiating events that nucleate full ring formation. As a result of these findings, we used OPTN translocation and ring formation as a positive identifier of depolarized mitochondria that were undergoing mitophagy for the remaining experiments.

## OPTN-mediated mitophagy occurs preferentially in the somatodendritic compartment of hippocampal neurons

Previous observations of mitophagy in neurons have reached differing conclusions on whether mitophagy occurs locally, for example at the site of mitochondrial damage in the axon, or instead is restricted to the somatodendritic compartment (*Ashrafi et al., 2014*; *Cai et al., 2012*; *Devireddy et al., 2015*; *Sung et al., 2016*; *Zheng et al., 2019*). To address this question, we first characterized the effect of mild oxidative stress in the axon. We measured axonal mitochondria and autophagosome dynamics and observed no differences in motility, density, and area flux between conditions (*Figure 3—figure supplement 1A–H*). Recent work has shown that oxygen/glucose deprivation and reperfusion induce a transient upregulation of retrograde mitochondrial motility (*Zheng et al., 2019*), so it possible that there was a transient response prior to our observation window. While we did note a decrease in the length of axonal mitochondria in both AO-free and AA conditions compared to control conditions, we found no significant difference in TMRE signal per cell (*Figure 3—figure supplement 1I–J*). The decrease in length suggests that mitochondrial fragmentation may be induced by oxidative damage, so we measured the average number of fusion and fission events per neuron, but saw no change in mitochondrial fission (*Figure 3—figure supplement 1K*).

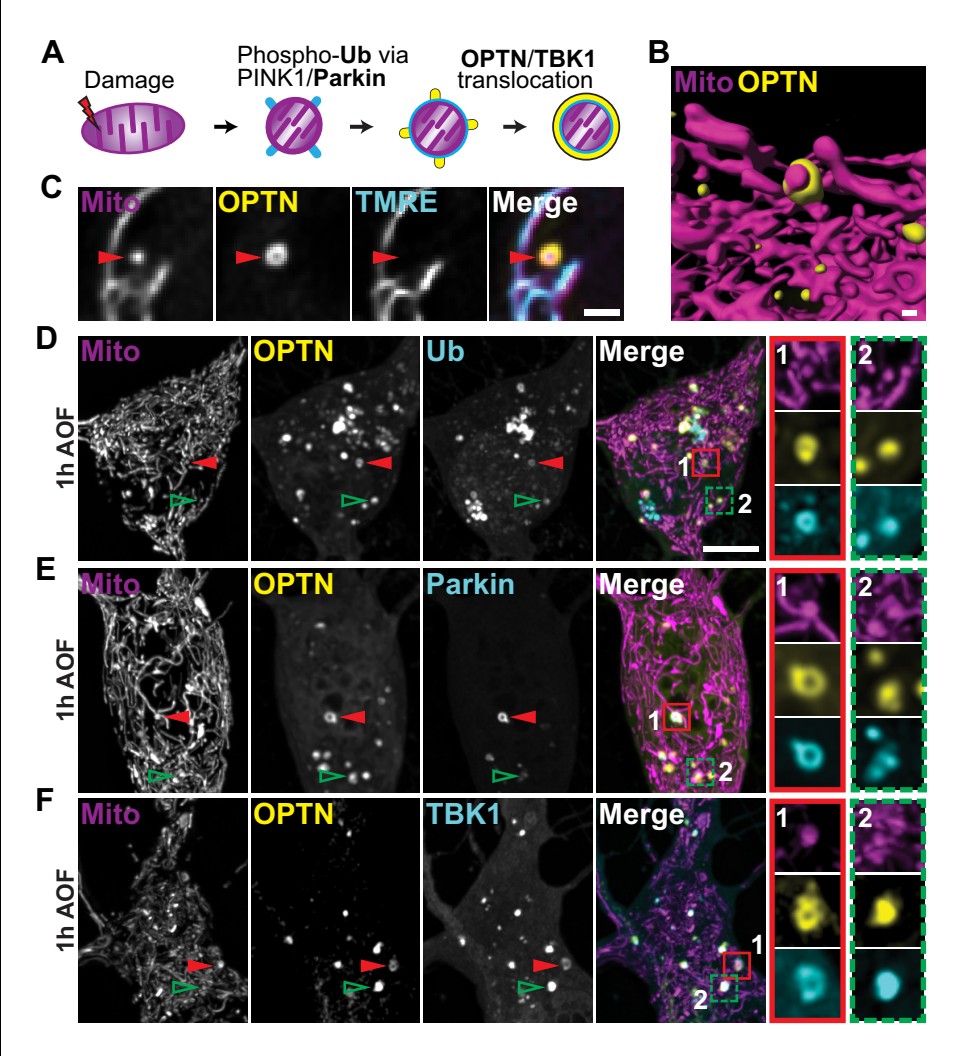

**Figure 2.** OPTN colocalizes with upstream mitophagy-associated proteins within an hour of antioxidant removal. (A) Schematic of the translocation of mitophagy-associated proteins to a damaged organelle. (B) Volume rendering of an OPTN-positive mitochondrion. Scale bar, 0.5 μm. (C) Representative image of an OPTN-positive mitochondrion that is TMRE-negative, confirming specific recruitment to damaged organelles. Scale bar, 1 μm. (D–F) Representative somal images of neurons expressing markers for mitochondria, OPTN, and ubiquitin (Ub; D), Parkin (E), and TBK1 (F). After 1 h in AO-free media, translocated proteins form rings around damaged mitochondria (1; red arrows and inset) and form puncta that colocalize with damaged mitochondria (2; green arrows and inset). Scale bar, 5 μm.

The online version of this article includes the following figure supplement(s) for figure 2:

**Figure supplement 1.** OPTN is expressed in various brain regions and localized throughout the neuron.
**Figure supplement 2.** OPTN fails to engulf damaged mitochondria when Parkin is depleted.

---

Next, we investigated the spatial dynamics of OPTN-mediated neuronal mitophagy by acquiring z-stacks of neuronal compartments (soma, dendrite, or axon) following mitophagy treatments (*Figure 3A–C*). Cells were transfected with Mito-DsRed, Halo-OPTN, and YFP-Nav$_{II-III}$, which localizes to the proximal region of the axon initial segment (AIS) and was used to discriminate between axons and dendrites. A higher percentage of AO-free treated neurons displayed OPTN-positive mitochondria compared to control cells (*Figure 3D*). Strikingly, in AO-free treated neurons ~ 95% of all mitophagic events occurred in the somal compartment, with limited events in either dendrites or axons (*Figure 3E*); this trend was also observed in control conditions. Given the much greater mitochondrial volume in the soma relative to the axon (somal mitochondrial content ~0.2 μm³/μm³; axonal

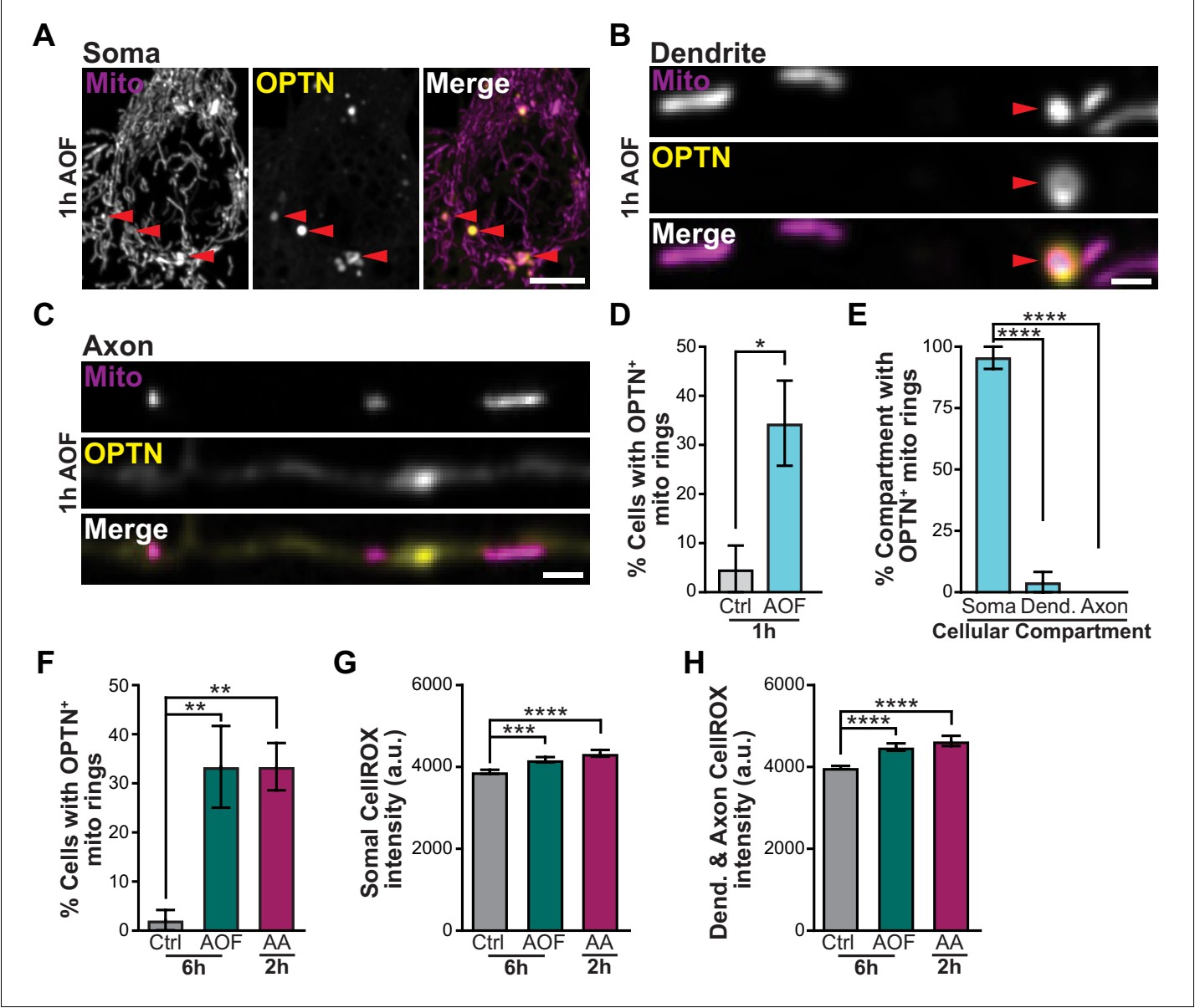

**Figure 3.** OPTN is specifically recruited to damaged mitochondria in the soma after mitophagy induction. (A–C) Representative image of the soma (A), dendrite (B), and axon (C) of a hippocampal neuron treated for 1 h with AO-free media. Mitophagy events are shown with red arrows. Scale bars: soma, 5 μm; axon and dendrite, 1 μm. (D) Quantification of the percent of neurons that contain OPTN-positive mitochondria. Mean ± SEM; n = 21-28 neurons from 3 biological replicates; 7 DIV. *, p < 0.05 by unpaired t test. (E) Quantification of the percent of OPTN-positive mitochondria in each cellular compartment for neurons treated for 1 h with AO-free media. Mean ± SEM; n = 21-28 neurons from 3 biological replicates; 7 DIV. ****, p < 0.0001 by one-way ANOVA with Dunn's multiple comparisons test. (F) Quantification of the percent of cells with OPTN-positive mitochondria rings for longer and harsher treatment conditions. Mean ± SEM; n = 36-48 neurons from 3-4 biological replicates; 7 DIV. **, p < 0.01 by one-way ANOVA with Dunn's multiple comparisons test. (G–H) Quantification of CellROX fluorescence intensity in the soma (G) and dendrites and axons (H). Mean ± SEM; n = 38-44 images from 3 biological replicates; 7-9 DIV. ***, p < 0.001; ****, p < 0.0001 by Kruskal-Wallis ANOVA with Dunn's multiple comparisons test. Dendrite, Dend.

The online version of this article includes the following figure supplement(s) for figure 3:

**Figure supplement 1.** Antioxidant removal alters axonal mitochondrial length with no effect on the TMRE fluorescence intensity.

mitochondrial content ~0.02 μm³/μm³), it is possible that these experiments undersampled axonal events. To address this possibility, we searched specifically for mitophagic events in the axons of control and treated cells. We found a very low frequency of OPTN-positive mitochondrial fragments

in axons that was not different between control and treated neurons (3/27 axons in control cells; 1/ 27 axons in AO-treated cells; 1/28 axons in AA-treated cells).

To follow up on this point, we used longer (6 h AO-free) and harsher (2 h AA) treatments to induce mitophagy and again observed similar levels of OPTN-positive mitochondria in the soma, but rarely in the axon (*Figure 3F*). To test whether our mitophagy inducing protocol preferentially targeted somal mitochondria as compared to axonal and dendritic mitochondria, we visualized Cell-ROX in all compartments. We observed similar levels of ROS generation in the soma and processes of treated neurons compared to control (*Figure 3G–H*). Consistently, there was change in the TMRE intensity of mitochondria in all compartments, implying that the mitochondrial network health was intact (*Figure 1C–D* and *Figure 3—figure supplement 1J*). We did note that analysis at the level of individual mitochondria revealed a minor decrease in the TMRE fluorescence intensity of axonal mitochondria following 2 h AA treatment compared to 6 h control conditions; this was not observed with 6 h AO-free treatments. We hypothesize that either long-term exposure to low doses (>2 hr) or short periods of high concentration of AA (*Ashrafi et al., 2014*) may disrupt mitochondrial bioenergetics causing axonal mitochondria to undergo mitophagy locally to preserve the health of the neuron. However, under conditions of mild oxidative stress, we find that neuronal mitophagy is generally restricted to the soma.

## Autophagosome engulfment occurs after OPTN ring formation

The multiple genetic links between components of the mitophagy pathway and neurodegeneration strongly suggest that failure to remove damaged organelles is detrimental to neuronal health. Therefore, the time course of removal is likely to be critical. In non-neuronal cell lines, the full engulfment of damaged mitochondria by LC3-positive autophagosomes can occur within an hour of an initiating insult, such as CCCP treatment or induction of localized ROS production via mito-KillerRed (*Moore and Holzbaur, 2016*; *Wong and Holzbaur, 2014a*). A similar time course was reported for sequestration and lysosomal removal of damaged axonal mitochondria (*Ashrafi et al., 2014*). However, another study found the pathway was just initiating 24 h post-CCCP treatment, with only a small fraction of neurons displaying detectable Parkin translocation to damaged organelles (*Cai et al., 2012*). We addressed this uncertainty by examining the time course for LC3 recruitment to damaged mitochondria following mitophagy induction (1 or 6 h AO-free or 2 h AA). Cells were transfected with Mito-DsRed, Halo-OPTN, and EGFP-LC3 (*Figure 4A*); in some experiments YFP-Nav$_{II-III}$ was used to discriminate between compartments, so neurons were instead transfected with Mito-SNAP, Halo-OPTN, and mScarlet-LC3 (*Figure 4B–C*). We visualized fragmented mitochondrial puncta that were OPTN- and LC3-positive in the soma, with limited events observed in the dendrites and axons, 6 h after antioxidant deprivation (*Figure 4A*; red arrows and inset).

LC3 ring formation was captured at multiple stages, as shown by the gallery of images (*Figure 4D*). Initially, fully formed OPTN rings around damaged mitochondria were LC3-negative. LC3 targeted to ubiquitinated OPTN-positive mitochondria appeared as a punctum on the OPTN ring, that expanded with time to form full LC3 rings surrounding mitochondrial fragments or mitophagosomes (*Figure 4D*); note that rings were observed in a single confocal plane, but these correspond to spheres in 3D reconstructions from z-stacks. We quantified the relative percentage of OPTN-positive rings that were LC3-negative, partially positive (a punctum or partial ring), or completely engulfed to determine whether there was a difference in the distribution among the experimental conditions. For all treatments, approximately a quarter of events were OPTN-positive but LC3-negative, a quarter displayed LC3 puncta or partially formed rings, and half of fragmented mitochondria were both OPTN- and LC3-positive (*Figure 4E*). These observations indicate that effective mitophagosome formation occurs within an hour after initial damage. This efficient sequestration of damaged organelles may be critical to prevent dissemination of damaged components throughout the mitochondrial network.

## Acidification of OPTN-positive mitochondria is a rate-limiting step in neuronal mitophagy

We next investigated the time course for lysosomal engulfment of damaged mitochondria by monitoring the recruitment of LAMP1, a late endosomal/lysosomal marker. Neurons were transfected with Mito-DsRed, Halo-OPTN, and LAMP1-EGFP and mitophagy was induced. We identified OPTN-

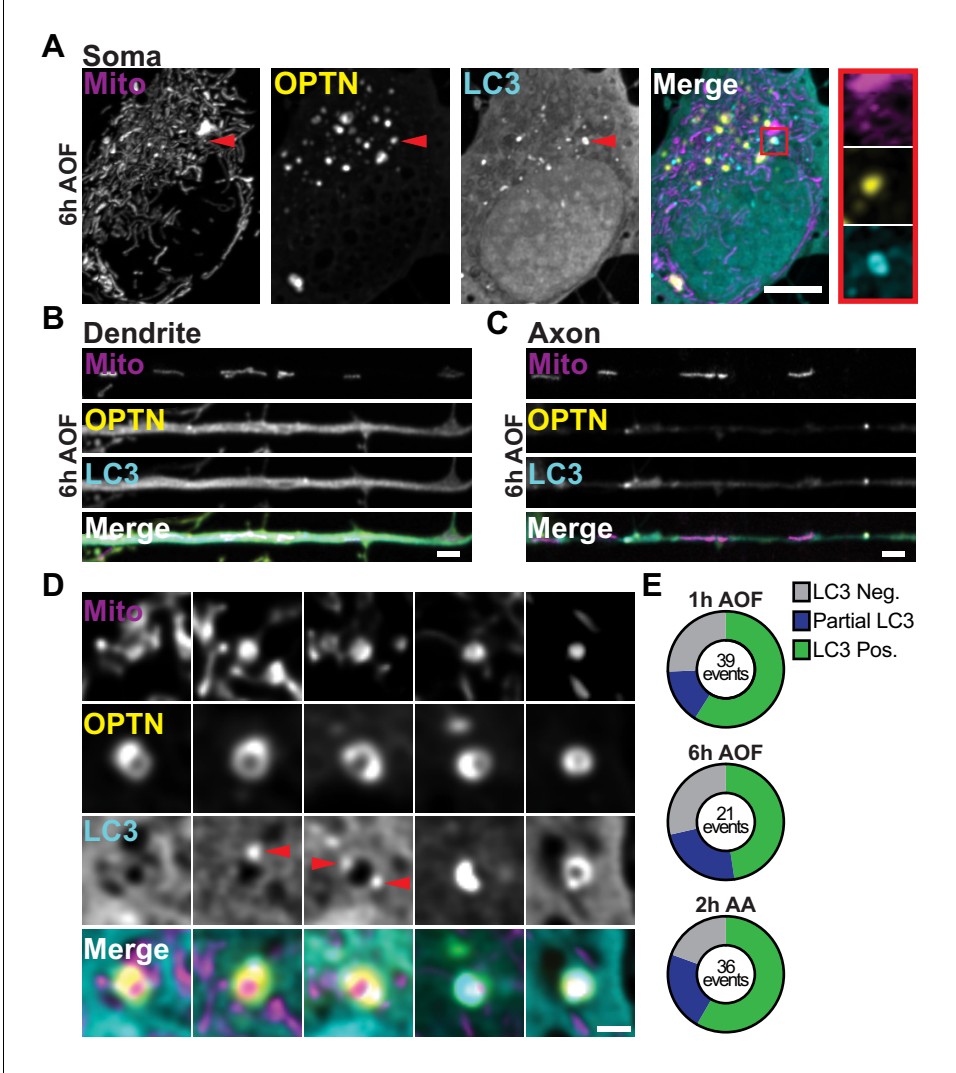

**Figure 4.** LC3 translocates to damaged mitochondria after OPTN ring formation. (**A–C**) Representative image of the soma (**A**), dendrite (**B**), and axon (**C**) of a hippocampal neuron treated for 6 h with AO-free media. A mitophagosome is specified by red arrows and inset. Scale bars: soma, 5 µm; axon and dendrite, 2 µm. (**D**) Gallery of representative images demonstrating fully formed OPTN rings around damaged mitochondria with various stages of LC3 ring development, including EGFP-LC3 positive, negative, or incomplete (containing a punctum or partial ring). Red arrows highlight LC3 puncta. Scale bar, 1 µm. (**E**) Quantification of the number OPTN rings that are LC3-positive, LC3-negative, or partial-LC3. The total number of events are listed for each condition. *n* = 36–48 neurons from 3 to 4 biological replicates; 7 DIV.

positive mitochondria and quantified the ratio of LAMP1-positive or LAMP1-negative events (*Figure 5A–B*). In control conditions, the vast majority of events were LAMP1-positive. Induction of mitophagy significantly increased the number of LAMP1-negative mitophagic events compared to control (*Figure 5B*). Thus, under basal conditions, neurons can efficiently sequester dysfunctional mitochondria, but perturbations to the system that increase organelle damage may overwhelm the pathway and stall lysosomal fusion.

A similar analysis was performed for neurons treated with LysoTracker Green (LysoT), a fluorescent dye that marks acidic vesicular compartments, to monitor the organelle acidification that is required for mitochondrial degradation (*Figure 5C–D*). In sharp contrast to our observations of LAMP1 co-localization, nearly all mitophagy events observed were LysoT-negative, no matter the treatment condition (*Figure 5D*). A small number of OPTN-positive mitochondria became LysoT-

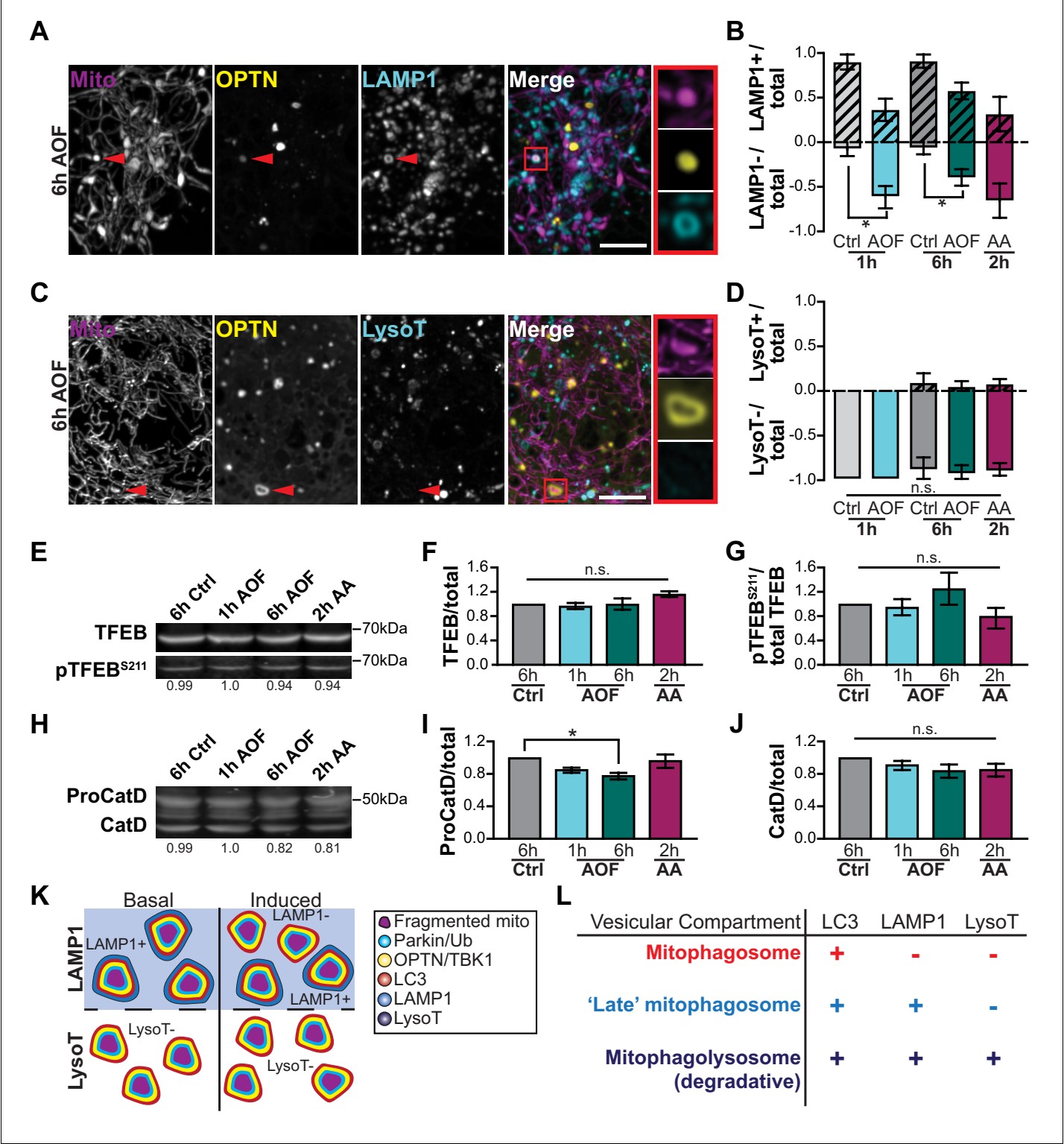

**Figure 5.** Mitochondria are sequestered in LAMP1-positive organelles that are non-acidified. (A) Representative image of a hippocampal neuron 6 h after AO-free treatment; an OPTN-LAMP1-positive event is indicated by the red arrows and the inset. Scale bar, 5 μm. (B) Quantification of the ratio of OPTN-positive mitochondria that are LAMP1-positive or LAMP1-negative compared to the total number of events. Mean ± SEM; *n* = 32-35 neurons from 3 biological replicates; 6-8 DIV. *, *p* < 0.05 by unpaired t test. (C) Representative image of a hippocampal neuron; an OPTN-positive mitochondrion is negative for Lysotracker Green (LysoT) 6 h after mitophagy induction (red arrows and inset). Scale bar, 5 μm. (D) Quantification of the ratio of OPTN-positive events that are LysoT-positive or LysoT-negative compared to the total number of events. Mean ± SEM; *n* = 33-40 neurons from

*Figure 5 continued on next page*

Figure 5 continued

3-4 biological replicates; 6-8 DIV. Not significant (n.s.) by one-way ANOVA with Tukey's multiple comparisons test. (E–G) Representative Western blot (E) and quantification of TFEB (F) and pTFEB[S211] (G) from cultured hippocampal neurons. Data shown as the fold change over control of TFEB divided by total protein stain (F) or as pTFEB[S211] over total TFEB (G). Normalization factors are shown under representative images. Mean ± SEM; n = 3 biological replicates; 7-8 DIV. Not significant (n.s.) by one-way ANOVA with Dunn's multiple comparisons test. (H–J) Representative Western blot (H) and quantification of ProCathepsin D (ProCatD; I) and mature Cathepsin D (CatD; J) from primary hippocampal neurons. Data shown as the fold change over control of the protein of interest divided by total protein stain. Mean ± SEM; n = 3 biological replicates; 7-8 DIV. Not significant (n.s.); *, $p < 0.05$ by one-way ANOVA with Dunn's multiple comparisons test. (K) Schematic of lysosomal fusion (marked by LAMP1) and acidification (marked by LysoT) of damaged mitochondria in basal and induced conditions. The majority of mitophagosomes are LAMP1-positive in control conditions, but only half of mitophagosomes are LAMP1-positive in treated neurons. However, most fragmented mitochondria are LysoT-negative in either basal or induced neurons. (L) Table depicting the various vesicular compartments that sequester damaged mitochondria.

The online version of this article includes the following figure supplement(s) for figure 5:

**Figure supplement 1.** Mitophagy induction does not alter the acidification of cellular autophagosomes or lysosomes.
**Figure supplement 2.** Mitophagosomes are quickly acidified to clear damaged mitochondria in HeLa cells.

positive at longer timepoints. These observations illustrate that following induction of neuronal mitophagy, acidification and turnover of damaged mitochondria is quite slow.

However, it may be that our mitophagy induction paradigm unexpectedly disrupts the acidification state of cellular autophagosomes and lysosomes. We examined the acidification of both organelles and determined that neither autophagosome nor lysosome acidification was disrupted by antioxidant deprivation (*Figure 5—figure supplement 1A–C*). These data are consistent with previous reports that autophagosomes effectively fuse with lysosomes to degrade cargo (*Boland et al., 2008*; *Kimura et al., 2008*; *Lee et al., 2011*; *Maday et al., 2012*). In addition, we monitored expression level of Transcription Factor EB (TFEB), a master transcriptional regulator of genes that are involved in lysosomal biogenesis and function (*Bajaj et al., 2019*; *Sardiello, 2016*). TFEB phosphorylation (pTFEB) at S211 sequesters the protein in the cytosol (*Martina et al., 2012*; *Roczniak-Ferguson et al., 2012*). We saw no changes in the levels of TFEB or pTFEB[S211] across all treatments compared to control (*Figure 5E–G*). We also examined Cathepsin D (CatD), a protease localized to lysosomes that nonspecifically degrades autophagosomal contents following autophagosome-lysosome fusion (*Benes et al., 2008*). ProCatD is targeted to intracellular vesicles where it undergoes processing to form an active enzyme, CatD (*Benes et al., 2008*). The levels of ProCatD remained consistent across treatments, with the exception of a minor decrease in expression following 6 h AO-free treatment (*Figure 5H–I*). No changes in expression levels of active CatD were seen in either the AO-free or AA conditions compared to control (*Figure 5H and J*). Thus, the delay in the degradation of damaged mitochondria reported here is specific to mitophagic events.

We compared these observations to the time course of mitophagy in HeLa cells treated with AA/Oligomycin A (10 μm) to initiate complete turnover of the mitochondrial network. We found a similar time course of mitochondrial fragmentation and engulfment by LC3, complete within ~1–2 hr. However, the time course of mitophagosome acidification was markedly faster in HeLa cells as compared to primary neurons, with >80% of OPTN-positive mitochondrial fragments also positive for LysoT within 1 h after mitophagy induction (*Figure 5—figure supplement 2*). Thus, we find that in neurons damaged mitochondria are efficiently sequestered, but the acidification of these mitophagosomes, which is required to fully degrade the engulfed mitochondrial fragments, takes more than 6 h and is much slower than the corresponding time course in HeLa cells expressing exogenous Parkin.

While LAMP1 is routinely used as a marker of lysosomes, recent work has demonstrated that a portion of LAMP1-positive organelles did not colocalize with lysosomal hydrolases, implying not all LAMP1 is on degradation-competent vesicles but also on late endosomes (*Cheng et al., 2018*; *Yap et al., 2018*). Under our conditions, we determined ~86% of LAMP1 or LAMP2 structures were LysoT-positive (*Figure 5—figure supplement 1D–E*). Since earlier steps in the formation of autophagosomes are not rate-limiting (*Figure 5—figure supplement 1F–G*), most damaged organelles are efficiently sequestered in autophagosomes within an hour of damage, forming mitophagosomes (*Figure 4E*). In control conditions, the majority of mitophagosomes were LAMP1-positive, but ≤50% were LAMP1-positive in treated conditions. Moreover, most mitophagosomes were LysoT-negative across all conditions, consistent with the slow acidification of engulfed mitochondria under both control and oxidative stress conditions (*Figures 5B, D and K*). These findings suggest that most

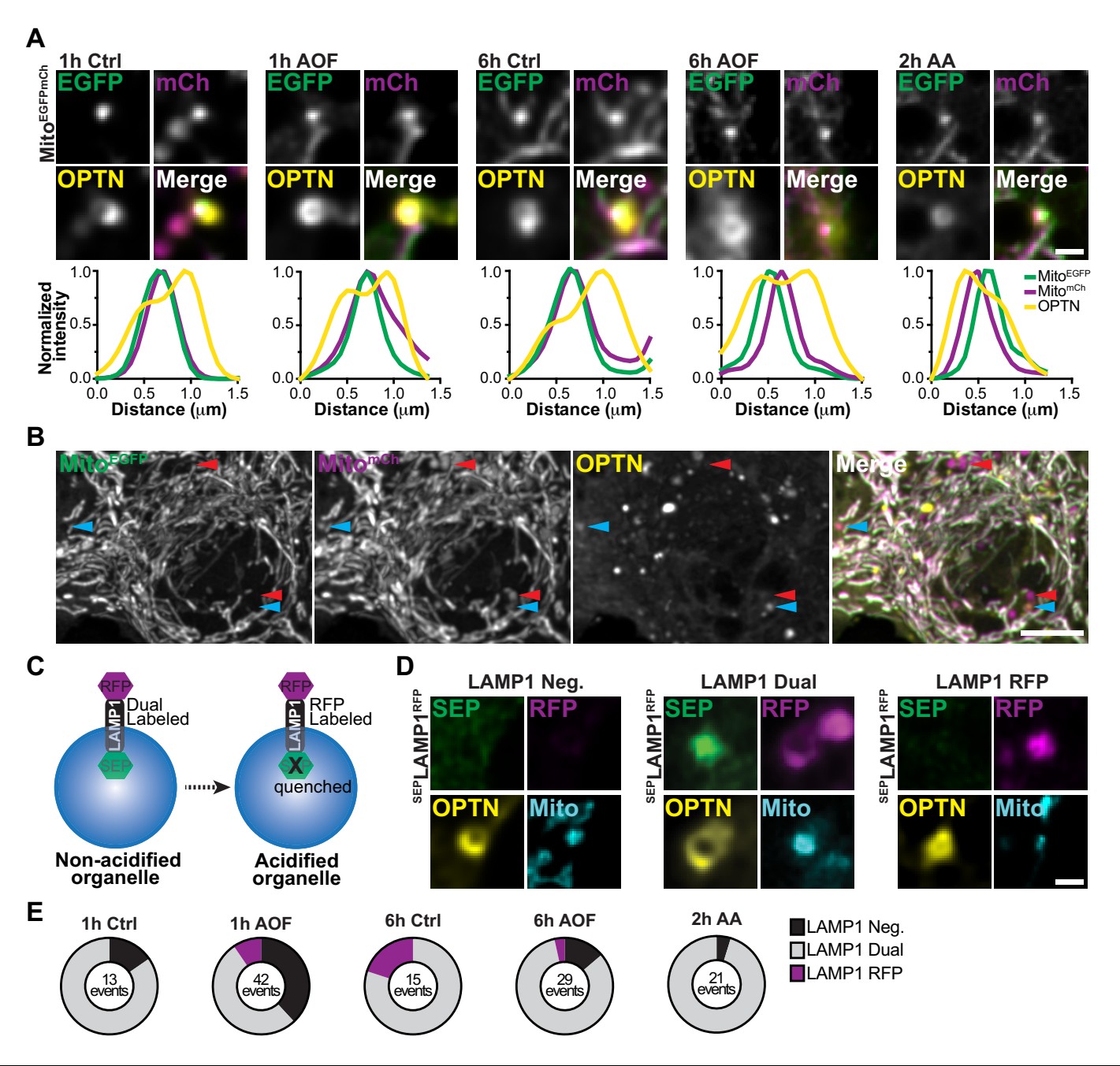

**Figure 6.** Lysosomal acidification is a rate-limiting step in the turnover of damaged mitochondria in neurons. (**A**) Representative images of OPTN-positive mitochondria that are labeled with a tandem Mito-EGFP-mCherry marker. Line scans of the mitophagy events are shown under representative images for all conditions. Scale bar, 1 μm. (**B**) Representative image of the dual labeled somal mitochondria network labeled with Mito-EGFP-mCherry. A population have undergone lysosomal fusion and acidification (shown as magenta only puncta; red arrows). A subset of this population is also OPTN-positive demonstrating turnover (blue arrows). Scale bar, 5 μm. (**C**) Schematic of the tandem SEP-LAMP1-RFP marker. In acidified environments, SEP is quenched and lysosomes are labeled only by RFP. (**D**) Representative images of OPTN-positive mitochondria showing conditions where they were LAMP1-negative, dual labeled in 'late' mitophagosomes, or RFP-only labeled in mitophagolysosomes. Scale bar, 1 μm. (**E**) Quantification of the number OPTN-positive mitochondria that are LAMP1-negative, LAMP1 dual labeled, or only labeled by RFP. The total number of events are listed for each condition. *n* = 30–32 neurons from 3 biological replicates; 6–7 DIV.

The online version of this article includes the following figure supplement(s) for figure 6:

**Figure supplement 1.** OPTN-positive mitochondria rings and puncta undergo lysosomal engulfment.

damaged mitochondria are efficiently engulfed in mitophagosomes (LC3-positive) or 'late' mitopha-gosomes (both LC3- and LAMP1-positive), but have yet to fuse with protease-containing lysosomes to become degradation-competent mitophagolysosomes (LC3-LAMP1-LysoT-positive; *Figure 5L*).

To test this idea, we examined mitochondrial flux using a tandem Cox8-EGFP-mCherry reporter. The fluorescence signal of EGFP, but not mCherry, quenches in acidic environments and thus the shift in fluorescence from yellow to red is commonly used to measure the efficiency of mitophagy (*Allen et al., 2013*; *Lee et al., 2018*; *McWilliams et al., 2016*; *Rojansky et al., 2016*). Under all con-ditions, we routinely identified OPTN-positive mitochondria that were dual labeled with EGFP and mCherry, illustrating these mitophagic events persist in non-acidified compartments representing mitophagosomes or 'late' mitophagosomes (*Figure 6A*). Distinct from the dual labeled mitochon-drial population, we identified some mCherry-only puncta (red arrows; *Figure 6B*). A subpopulation of these puncta colocalized with OPTN (blue arrows), suggesting mitochondrial turnover via mitophagy.

Since we observed differences in the localization of LAMP1 and LysoT with mitophagosomes, we performed similar experiments using a tandem SEP-LAMP1-RFP reporter, where supereclliptic pHluorin (SEP) is fused to the lumen of LAMP1-RFP (*Figure 6C*; *Farías et al., 2017*). Thus, we could visualize both LAMP1 localization and acidification, as the fluorescence signal of the SEP will quench in acidic environments while the RFP signal will persist (pH <6; *Sankaranarayanan et al., 2000*). The temporal dynamics of lysosomal fusion and acidification were determined by comparing LAMP1 recruitment to damaged mitochondria following mitophagy induction (1 or 6 h control/AO-free or 2 h AA). We quantified the relative percentage of OPTN-positive mitochondria rings that were LAMP1-negative, dual LAMP1 labeled, or LAMP1 RFP-only labeled (*Figure 6D–E*). Consistent with previous data, mitophagic events in control cells were predominantly LAMP1-positive. Interestingly, there was a shift in a sub-population of events that were LAMP1-negative after 1 h to LAMP1 RFP-only following 6 h in control media, exhibiting mitochondrial turnover on a longer time scale. Induc-tion of mitophagy by antioxidant deprivation substantially increased both the fraction of LAMP-1 negative and the total number of mitophagy events compared to control (*Figure 6E*). However, as with the tandem reporter for mitochondria, most mitophagic events in all conditions were dual labeled with SEP and RFP, confirming that these damaged organelles are sequestered in a non-acidi-fied 'late' mitophagosome.

As previously mentioned, two types of OPTN localization were observed with damaged mito-chondria, OPTN rings and puncta (*Figure 2D–F*). OPTN puncta may represent an early stage of nucleation, but it also conceivable that an OPTN ring collapses into punctum following rounds of lysosome fusion at the end stages of mitophagy (*Figure 6—figure supplement 1A*), causing the pre-ring and post-ring formation states of OPTN to be morphologically indistinguishable. To address this possibility, we repeated our quantification, this time including OPTN-positive mitochondrial puncta and rings. Using this criterion, we observed a larger percentage of LAMP1-negative and LAMP1 RFP-only mitophagic events, as well as an overall increase in the total number of events for all treatments (*Figure 6—figure supplement 1B*). This analysis suggests that OPTN-positive puncta are most likely to be in non-acidified organelles formed prior to the expansion of OPTN to form a full ring, and further support a model were sequestration of damaged organelles occurs quickly, but acidification is delayed in neuronal mitophagy.

## 'Aged' mitochondria are sequestered in the soma during basal mitophagy

To investigate long-term autolysosome sequestration hours after initial damage, we performed SNAP-tag pulse-chase experiments. Neurons were transfected with Mito-SNAP, Halo-OPTN, and in some experiments LAMP1-EGFP. The following day, mitophagy was initiated and mitochondria were labeled with the first SNAP ligand for 30 min. Subsequently, treated neurons were placed in control maintenance media, 2 h SNAP block was used to saturate the remaining SNAP-binding sites, and neurons were incubated overnight. Prior to imaging, cells were labeled with a second SNAP ligand (a spectrally distinct fluorophore) for 30 min and in some experiments labeled with LysoT (*Figure 7A*). As a result, 'age'-related mitochondrial sequestration and elimination can be examined since 'Old' (labeling of Mito-SNAP expressed from the time of transfection until SNAP Block label-ing;~24 hr) and 'Young' (labeling of Mito-SNAP expressed after SNAP Block until imaging;~22 hr) mitochondrial populations are distinct (*Figure 7B–C*). Using multiple lysosomal markers [*Figure 7B*;

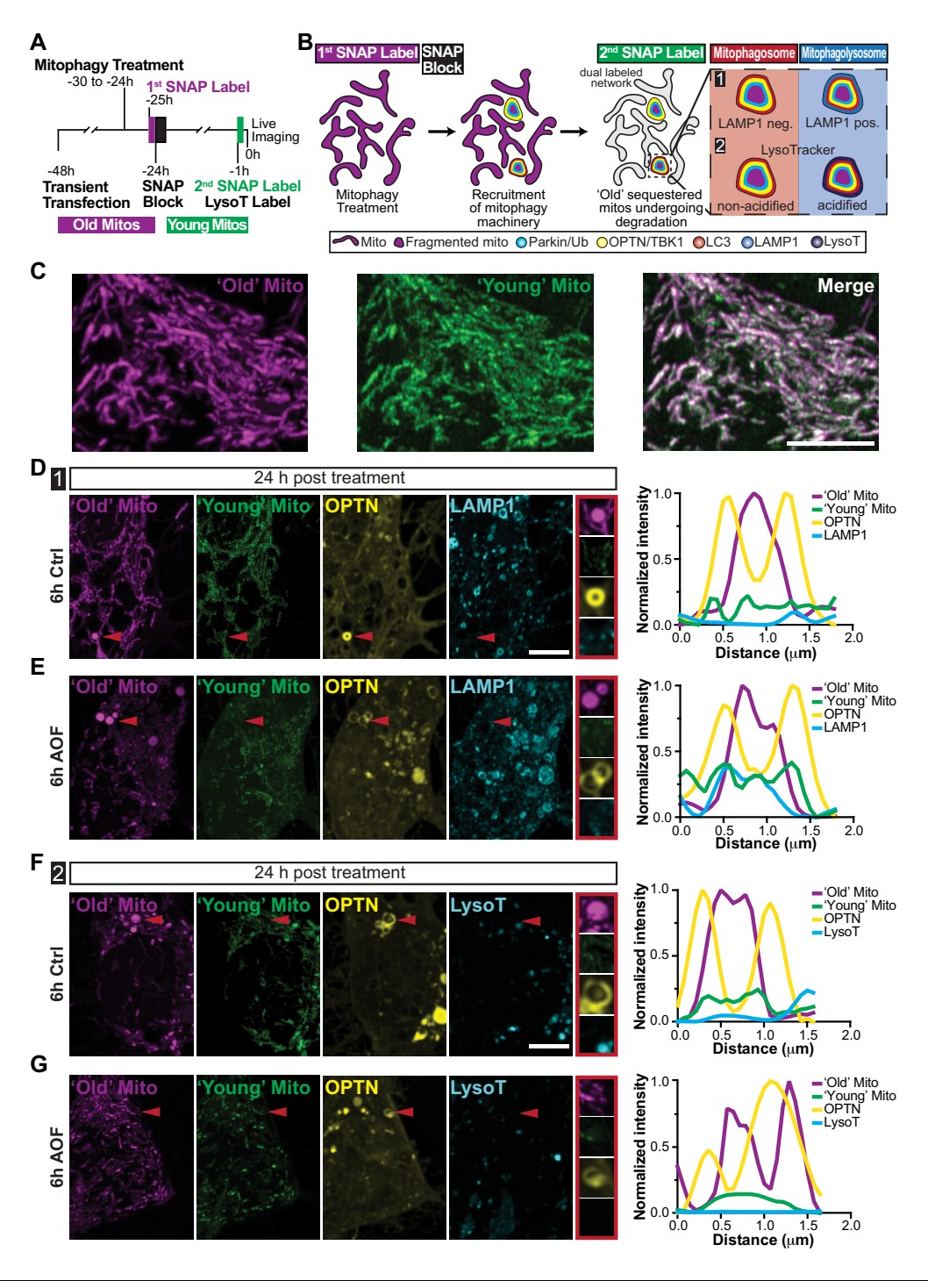

**Figure 7.** 'Aged' mitochondria are sequestered in non-acidified compartments twenty-four hours after treatment. (A–B) Time line (A) and schematic (B) of mitochondrial pulse-chase experiments. Following transient transfection, mitophagy was induced (6 h control or AO-free) and mitochondria were labeled with the first SNAP ligand ('Old' Mito; shown as magenta). SNAP block was added for 2 h to saturate the remaining binding sites and neurons were left overnight. Prior to imaging, mitochondria were labeled with a second spectrally distinct SNAP ligand ('Young' Mito; shown in green). Both 'Old' and 'Young' ligands were labeled for 30 min, followed by two quick washes, and a 30 min washout. (C) Representative image of 'Old' and 'Young' mitochondrial populations in the soma. Scale bar, 5 μm. (D–E) Representative images of neurons that are 24 h post-treatment illustrating OPTN sequestered 'Old' mitochondria that are negative for LAMP1 and the 'Young' mitochondria marker (red arrows

*Figure 7 continued on next page*

*Figure 7 continued*

and insets). Scale bar, 5 µm. Line scans of mitophagy events are shown on the right. (**F–G**) Representative somal images of neurons that are 24 h post-treatment. Red arrows highlight OPTN-positive 'Old' mitochondria that are negative for 'Young' mitochondria and LysoT. Scale bar, 5 µm. Line scans of mitophagic events are shown to the right of the image.

LAMP1 (1) and LysoT (2)], we determined whether OPTN-positive mitochondria were sequestered in mitophagosomes (red; LAMP1- and LysoT-negative) or in mitophagolysosomes (blue; LAMP1- and LysoT-positive).

Twenty-four hours post-treatment, we visualized spherical, single labeled 'Old' Mito-SNAP puncta that were engulfed by Halo-OPTN rings. Strikingly, these OPTN-positive mitochondria were negative for 'Young' Mito-SNAP and LAMP1-EGFP or LysoT (*Figure 7D–G*), indicating these isolated 'Old' mitochondrial fragments were engulfed by OPTN within hours of initial damage, as they were not labeled by the 'Young' SNAP ligand, but still had not undergone autolysosome sequestration and degradation 24 h later. Not only were these delayed mitophagic events observed in the soma of AO-free treated neurons (LAMP1, 10.3%; LysoT, 4%; *Figure 7E and G*), they were also seen in control treated neurons (LAMP1, 6.3%; LysoT, 13.3%; *Figure 7D and F*), revealing this slow mitochondrial turnover is detected under both basal and induced conditions. Thus, while neurons can efficiently undergo mitophagosome and 'late' mitophagosome formation and sequestration, it appears there is inefficient turnover where the degradation of damaged mitochondria can occur >24 h after initial damage. Even under basal conditions, slow mitochondrial removal could cause a buildup of depolarized organelles, exposing the neuron to damage and disease-associated mitophagy mutants may exacerbate this pathology.

## Expression of an ALS-associated OPTN mutant increases the neurons susceptibility to stress and damage

Genetic studies have identified mutations in OPTN as causative for ALS and frontotemporal dementia (FTD; *Ito et al., 2011*; *Maruyama and Kawakami, 2013*; *Pottier et al., 2015*). OPTN$^{E478G}$ is a heterozygous missense mutation identified in both sporadic and familial cases of ALS (*Figure 8A*; *Maruyama and Kawakami, 2013*; *Maruyama et al., 2010*). In HeLa cells, OPTN$^{E478G}$ inhibits mitophagy by failing to translocate to damaged, ubiquitinated mitochondria (*Moore and Holzbaur, 2016*). OPTN$^{E478G}$ has also been linked to the activation of inflammation, a hallmark of ALS (*Gkikas et al., 2018*). We sought to test the functional significance of ALS-associated OPTN$^{E478G}$ expression on the mitophagy pathway by examining mitochondrial health in primary neurons expressing this mutation.

Endogenous OPTN was depleted using siRNA, leading to a ~ 60% reduction in OPTN protein levels (*Figure 8—figure supplement 1*). Expression of OPTN$^{E478G}$ did not grossly alter somal mitochondrial content, but did affect the morphology of the mitochondrial network, resulting in the appearance of swollen organelles, as compared to expression of OPTN$^{WT}$ (*Figure 8B–D*). These large mitochondria were observed in both basal and treated conditions, suggesting OPTN$^{E478G}$ expression is sufficient to alter mitochondrial morphology. In addition, there was a decrease in the number of OPTN$^{E478G}$ puncta compared to OPTN$^{WT}$ (*Figure 8E*) and mutant puncta did not colocalize with mitochondria as often. It is likely that OPTN$^{E478G}$ remains more cytosolic due to its inability to bind ubiquitin and associate with damaged mitochondria. As a result, OPTN$^{E478G}$ fails to initiate mitophagy and the turnover of depolarized mitochondria, leading to accumulation of swollen organelles.

Finally, we asked whether OPTN$^{E478G}$ expression altered mitochondrial network polarization. We saw a significant decrease in TMRE fluorescence intensity of mitochondria in neurons expressing OPTN$^{E478G}$ compared to OPTN$^{WT}$ in AO-free treated neurons, illustrating mitochondrial membrane potential was altered with mutant OPTN expression (*Figure 8F*). Surprisingly, this was also observed in control conditions, so OPTN$^{E478G}$ expression alone was sufficient to disrupt mitochondrial health. Furthermore, in an OPTN$^{E478G}$ background AO-free treatments exacerbated the effect compared to control. Together these findings suggest a model where in a heterozygous background, OPTN$^{E478G}$

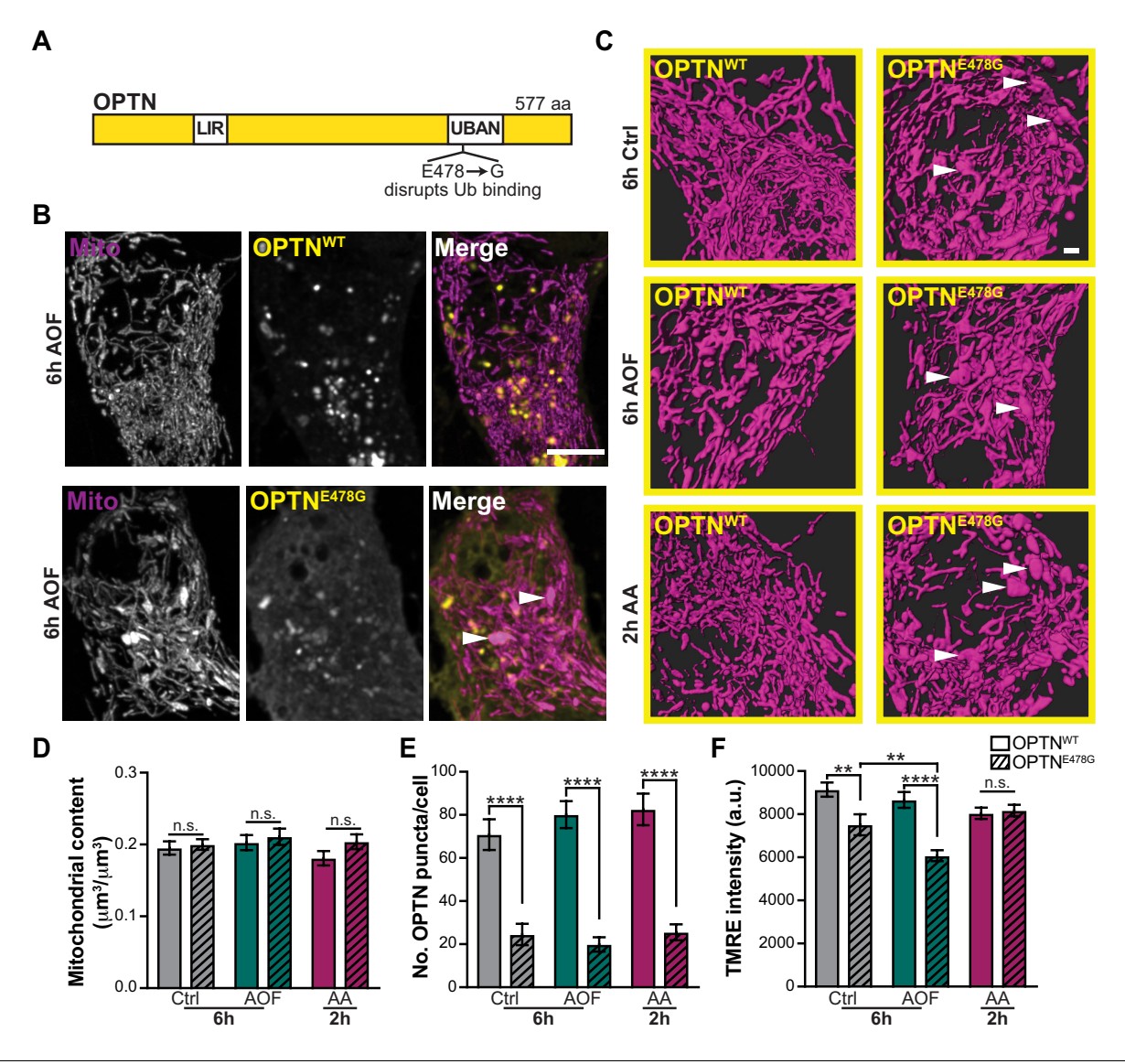

**Figure 8.** A disease-associated OPTN mutant increases mitochondrial vulnerability to oxidative stress. (**A**) Schematic of OPTN and its various domains. LIR, LC3 interacting region. UBAN, ubiquitin binding in ABIN and NEMO. The ALS-associated OPTN mutant E478G fails to bind ubiquitin. (**B**) Representative image of AO-free treated neurons expressing WT OPTN (OPTN^WT; upper panel) or a disease-linked OPTN mutant (OPTN^E478G; lower panel). White arrows denote swollen mitochondria. Scale bar, 5 μm. (**C**) Volume renderings of the somal mitochondrial network; OPTN^E478G expression induces the appearance of enlarged organelles compared to the expression of OPTN^WT (white arrows). Scale bar, 1 μm. (**D**) Quantification of the somal mitochondrial content. Mean ± SEM; $n$ = 24-40 neurons from 5 biological replicates; 8 DIV. Not significant (n.s.) by unpaired t test. (**E**) Quantification of the number of OPTN puncta per cell. Mean ± SEM; $n$ = 24-40 neurons from 5 biological replicates; 8 DIV. ****, $p < 0.0001$ by Kruskal-Wallis ANOVA with Dunn's multiple comparisons test. (**F**) Quantification of the TMRE fluorescence intensity. Mean ± SEM; $n$ = 24-40 neurons from 5 biological replicates; 8 DIV. **, $p < 0.01$; ****, $p < 0.0001$ by Kruskal-Wallis ANOVA with Dunn's multiple comparisons test.

The online version of this article includes the following figure supplement(s) for figure 8:

**Figure supplement 1.** OPTN is efficiently knocked-down in neurons with siRNA.

expression alters mitochondrial health but functional OPTN^WT expression maintains sufficient levels of mitophagy. However, in stressed conditions which further disrupts mitochondrial health, levels of mitophagy are no longer adequate to maintain the health of the neuron. Thus, expression of the ALS-linked OPTN^E478G mutant becomes sufficient to disrupt mitophagy and cause neurodegeneration.

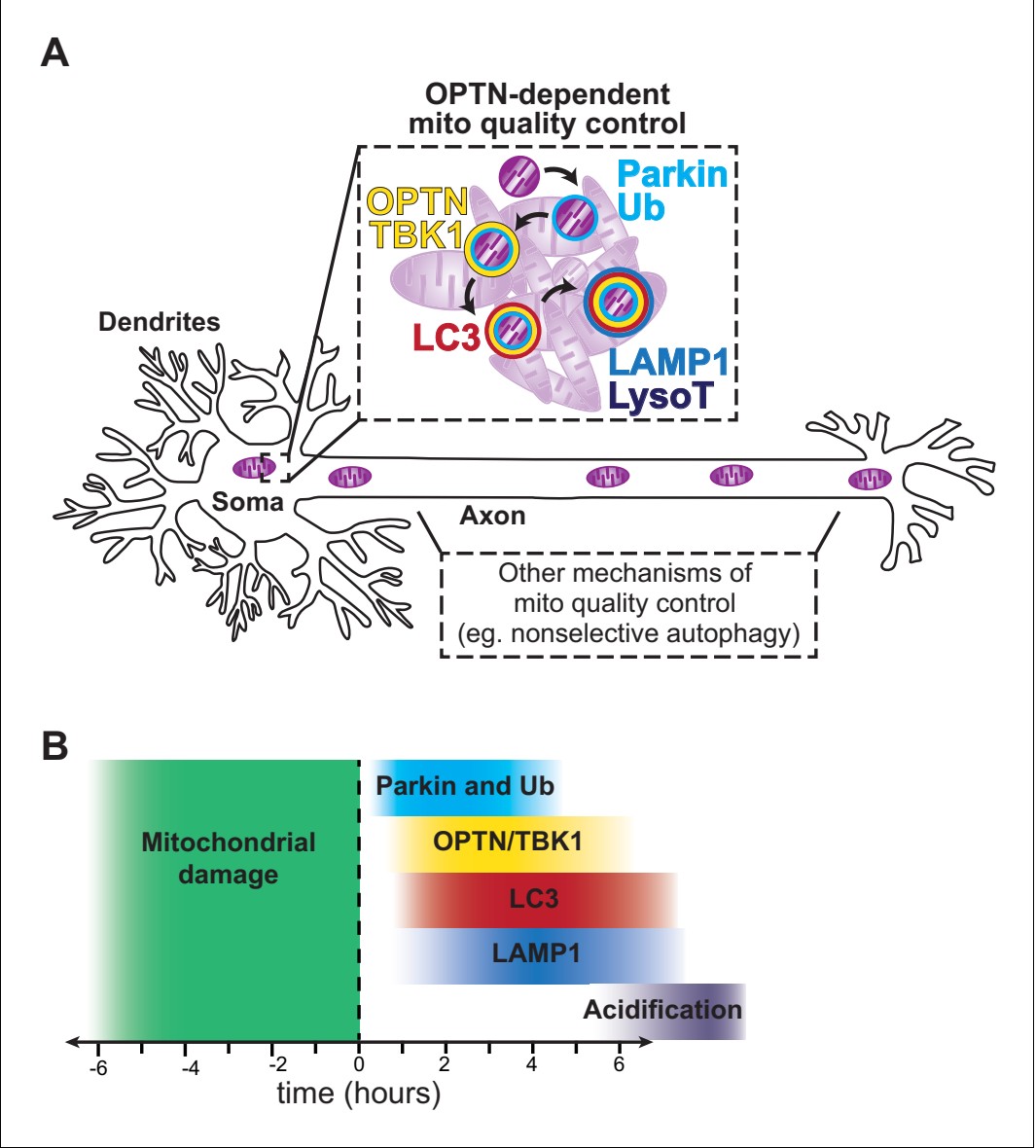

**Figure 9.** Model depicting the spatial and temporal regulation of OPTN-mediated neuronal mitophagy. (**A**) Upon mitophagy induction, Parkin translocates to spherical mitochondria and increases the abundance of ubiquitin chains. OPTN and its kinase TBK1 are recruited followed by sequestration and elimination via autophagosome engulfment and lysosomal fusion, as monitored by LC3 and LAMP1/LysoT, respectively. This quality control mechanism is compartmentally restricted to the soma and rarely occurs in the axon. As a result, other quality control pathways may regulate axonal mitochondria. (**B**) Parkin, Ub, TBK1 and OPTN localize with damaged organelles within an hour of inducing mitophagy. LC3 translocation occurs after OPTN and ~75% of OPTN-positive mitochondria are LC3-positive, forming mitophagosomes an hour after initial damage. Under basal conditions, 'late' mitophagosomes (OPTN-LAMP-positive mitochondria) routinely form within an hour. However, mitophagy induction perturbs this pathway, increasing the number of LAMP1-negative OPTN-positive mitochondria. Interestingly, only a small fraction of OPTN-positive mitochondria are acidified in either basal or induced conditions, suggesting lysosomal acidification to eliminate damaged organelles is rate-limiting.

## Discussion

The mitophagy pathway has been studied in detail in non-neuronal cells lines, leading to the identification of OPTN as an autophagy receptor that facilitates the engulfment of mitochondrial fragments in a PINK1/Parkin-dependent manner (*Lazarou et al., 2015*; *Moore and Holzbaur, 2016*; *Wong and Holzbaur, 2014a*). However, a rigorous analysis of OPTN-mediated mitophagy in

neurons has yet to be carried out, limiting our understanding of how this mechanism functions in neurons expressing endogenous levels of Parkin, and further, how defects in the pathway might lead to neurodegeneration. In this study, we used multicolor live-cell imaging to examine the spatiotemporal dynamics of OPTN-mediated mitophagy in primary hippocampal neurons. We find that mild oxidative stress, induced through antioxidant deprivation or by low doses of AA, results in minor somal and axonal mitochondrial fragmentation, without compromising the overall network (*Figure 1C–F* and *Figure 3—figure supplement 1*). We noted that OPTN was dynamically recruited to damaged TMRE-negative organelles following these treatments, indicating initiation of the selective mitophagic pathway (*Figure 2C*). Using this experimental paradigm, we saw frequent mitophagic events in the soma, with only limited events in either dendrites or axons (*Figure 3*). This finding demonstrates that OPTN-dependent mitophagy occurs in a spatially-restricted manner in primary neurons (*Figure 9A*).

Additionally, we determined that 'late' mitophagosome formation is efficient under basal conditions, but oxidative stress can overwhelm the system. Surprisingly, lysosomal acidification and turnover were rarely observed in either basal or stressed conditions at all time points tested (*Figure 5*). These findings were confirmed using tandem markers for mitochondria and lysosomes, where we could simultaneously monitor localization and acidification (*Figure 6* and *Figure 6—figure supplement 1*). Moreover, mitophagosomes that were LAMP1- and LysoT-negative were still visualized a full day after treatment (*Figure 7*). Taken together, these results illustrate that sequestered mitochondria are persistently sequestered in non-acidified organelles, where removal of damaged mitochondria is slow and inefficient (*Figure 9B*). Interestingly, this delay was shown to be specific to these mitophagic events, as most neuronal lysosomes were efficiently acidified (*Figure 5—figure supplement 1*). We speculate that this slow acidification is attributed to the multiple autophagosome-lysosome fusion events that may be required to fully acidify a mitophagolysosome. Alternatively, ion transporters or other proteins associated with this vesicular compartment could be differentially regulated (*Majumdar et al., 2011*), but further work will be required to determine the underlying cause. Of note, the slow time course of mitophagosome acidification in primary neurons following very limited mitochondrial damage differs markedly from the more rapid process in HeLa cells engaged in the wholesale turnover of their mitochondrial networks (*Figure 5—figure supplement 2*).

Importantly, the slow removal of sequestered mitochondrial fragments occurs under both basal and stressed conditions, suggesting that this is a potential point of vulnerability in the pathway supporting neuronal homeostasis. This susceptibility can be exacerbated by multiple factors related to aging and neurodegeneration, including increased levels of oxidative damage to cellular organelles, changes in lysosomal acidification with aging (*Lee et al., 2011*; *Lie and Nixon, 2019*), or the expression of mutations in pathway components. These alterations to the system might further impede the rate of turnover, increasing neuronal sensitivity to damage and degeneration. We tested this idea by expressing an ALS-linked OPTN mutation, and observed an overall decrease in the membrane potential of the mitochondrial network (*Figure 8F*). It is likely that damaged mitochondria were not efficiently sequestered but instead free to fuse with the cellular network disrupting overall cellular health. We propose that expression of mutant OPTN is detrimental to the neuron through a feedforward mechanism. First, OPTN$^{E478G}$ fails to translocate to damaged organelles, which may perturb the mitophagy pathway in a dominant-negative manner (*Moore and Holzbaur, 2016*). Second, decreased mitochondrial network health due to mutant expression could increase the number of damaged organelles that need to be recycled via mitophagy. However, OPTN$^{E478G}$ is not recruited to these depolarized mitochondria, further compounding the problem, leading to cellular stress and potential neurodegeneration.

Since the OPTN$^{E478G}$ mutation is autosomal dominant, one copy is sufficient to cause neurodegeneration and ALS (*Maruyama et al., 2010*). In an OPTN$^{WT}$ background, our data indicates that the mitophagic pathway can cope with hours of oxidative stress to effectively sequester damaged mitochondria and maintain neuronal health. However, the expression of OPTN$^{E478G}$ is sufficient to disrupt the mitochondrial network even under control conditions. In this mutant background, a second hit to the system induced by long-term oxidative stress resulted in additional mitochondrial network damage (*Figure 8F*). It appears that mutant expression and stressed conditions further enhance the susceptibility of the neuron to cellular stress. Thus, we propose that mitochondrial quality control is tightly regulated in neurons, where sequestration of damaged organelles is critical to

prevent dissemination of damaged components and disruption of the pathway is sufficient to cause neurodegeneration.

A constitutive pathway for axonal autophagy has been reported in neurons both in vitro (*Maday and Holzbaur, 2014*; *Wong and Holzbaur, 2014b*) and in vivo (*Neisch et al., 2017*; *Soukup et al., 2016*; *Stavoe et al., 2016*). In this pathway, autophagosomes non-selectively engulf and degrade mitochondrial fragments at the axon tip. But this is distinct from the selective receptor-mediated clearance of damaged mitochondria described here, which is almost entirely a somal quality control mechanism. The spatial specificity of mitophagy was previously postulated based on observations in vivo in *Drosophila* (*Devireddy et al., 2015*; *Sung et al., 2016*), qualitative findings of acidified mitochondria only within the soma in *mito*-QC mice (*McWilliams et al., 2016*), and visualization of the somal clearance of ischemically damaged axonal mitochondria (*Zheng et al., 2019*). Countering this literature is a report of axonal mitochondria turnover downstream from PINK1 and Parkin (*Ashrafi et al., 2014*) and studies indicating that axonal mitochondria motility is arrested following mitochondrial damage (*Hsieh et al., 2016*; *Liu et al., 2012*; *Wang et al., 2011*), necessitating a local clearance mechanism. It may be that under wholesale disruption of the mitochondrial network, initiated by CCCP or higher levels of Antimycin/Oligomycin, there is more widespread induction of mitophagy downstream from Parkin, leading to the local mitophagy of axonal mitochondria previously described (*Ashrafi et al., 2014*). However, we argue that the limited damage examined here may more accurately reflect the levels of mitochondrial turnover seen in vivo and during aging, stress, or expression of dominant mutations in pathway components, such as the OPTN$^{E478G}$ mutant. Under these conditions, as we demonstrate in vitro and was previously shown in vivo (*Devireddy et al., 2015*; *McWilliams et al., 2016*; *Sung et al., 2016*), the PINK1/Parkin mitophagy pathway is compartmentally restricted to the soma. Thus, neuronal mitophagy is both molecularly and spatially distinct from the previously characterized constitutive axonal autophagy pathway (*Evans and Holzbaur, 2019*).

While we focus here on the PINK1/Parkin pathway, given its genetic links to neurodegenerative diseases like PD and ALS, it is likely that additional and potentially compensatory pathways are actively contributing to mitochondrial quality control in neurons. Previous studies have proposed that mitochondrial ubiquitin ligase 1 (MUL1) plays a role in mitochondrial quality control in neurons, where it functions as an additional E3 ubiquitin ligase in a parallel pathway to Parkin (*Ambivero et al., 2014*; *Puri et al., 2019*; *Yun et al., 2014*). Other studies have suggested that alternative pathways such as NIPSNAP/Parkin-mediated mitophagy (*Nautiyal et al., 2010*; *Princely Abudu et al., 2019*) and NIX/BNIP3L-dependent mitophagy (*Bellot et al., 2009*; *Koentjoro et al., 2017*; *Real et al., 2005*; *Sowter et al., 2001*) support mitochondrial clearance. Further studies will be required to quantitatively assess the interplay of these various pathways in mitochondrial quality control in neurons and the extent of their overlap. In addition, further work will be required to determine the relative contributions of defects in mitophagy and systemic changes in inflammation that can also be induced by mutations in PINK1 and/or Parkin, although it is likely that both aspects contribute to the onset of neurodegeneration in PD and perhaps ALS. The continued development of more physiologically relevant model systems coupled to rigorous quantitative analysis will be required to address these important questions and advance therapeutic strategies for neurodegeneration.

## Materials and methods

### Reagents

Constructs used include the following: Mito-DsRed (kindly provided by. T. Schwartz, Harvard Medical School, Boston), Mito-SNAP (recloned from Mito-DsRed into a pSNAPf [New England Biolabs]), Cox8-EGFP-mCherry was subcloned from Addgene 78520, pEGFP-LC3B (a gift from T. Yoshimori, Osaka University, Osaka), mScarlet-LC3B (EGFP was replaced with mScarlet from Addgene 85054), mCherry-EGFP-LC3 (kindly provided by T. Johansen, University of Tromso, Tromso, Norway), pEGFP-OPTN (kindly provided from I. Dikic, Goethe University, Frankfurt), Halo-OPTN (subcloned from EGFP-OPTN to a pHaloTag vector [Promega]), Halo-OPTN$^{E478G}$ was subcloned from HA-OPTN$^{E479G}$ (kindly provided from I. Dikic, Goethe University, Frankfurt) to a pHaloTag vector (Promega), YFP-Parkin and mCherry-Parkin (a gift from R. Youle, NIH, Bethesda), mutant mCherry-Parkin T240R

was generated from site-directed mutagenesis, untagged-Parkin (mCherry-Parkin was removed), YFP-Nav$_{II-III}$ (Addgene 26056), GFP-Ub (Addgene 11928), TBK1 (Addgene 23851) was recloned into a pSNAPf vector (New England Biolabs), LAMP1-EGFP was subcloned from Addgene 1817 where RFP was replaced with EGFP, LAMP2-EGFP (provided by E. Chapman, University of Wisconsin-Madison was subcloned into pEGFP_N1), SEP-LAMP1-RFP (kindly provided by J. Bonifacino, NIH, Bethesda) and SNAP-WIPI2B (subcloned from GFP-WIPI2B). All constructs were verified by DNA sequencing. ON-TARGET*Plus* Rat OPTN (246294) siRNA *SMARTpool* (L-097177-02-0005) and ON-TARGET*Plus* Rat Prkn (56816) siRNA *SMARTpool* (L-090709-02-0005) were purchased from Dharmacon. Reagents used include the following: TMRE (tetramethylrhodamine ethyl ester, Ethyl Ester, Perchlorate; Life Technologies, T-669), Antimycin A (Sigma-Aldrich, A8674), Oligomycin A (Sigma Aldrich, 75351), LysoTracker Green DND-26 (ThermoFisher, L7526) and Deep Red (ThermoFisher, L12492), MitoTracker Green FM (ThermoFisher M7514), CellMask Orange Plasma membrane Stain (ThermoFisher 10045), B-27 Supplement minus antioxidants (ThermoFisher, 10889038), and CellROX Deep Red Reagent (Invitrogen, C10422). HaloTag constructs were labeled with Janelia Fluor 646 HaloTag (Promega, GA1120) and SNAP-Tag constructs were labeled with JF646-SNAP (provided by Luke Lavis, Janelia Farms), SNAP-Cell TMR-Star (New England Biolabs, S9015S), SNAP-Cell 430 (New England Biolabs, S9109S), and SNAP-Cell Block (New England Biolabs, S9106S). For a list of key reagents, see *Supplementary file 1*.

## Primary hippocampal culture

Sprague Dawley rat hippocampal neurons at embryonic day 18 were obtained from the Neurons R Us Culture Service Center at the University of Pennsylvania. Cells (fixed imaging, 175,000 cells; live imaging, 250,000 cells) were plated on 25 mm acid-washed glass coverslips (World Precision) or in a 35 mm glass-bottom dishes (MatTek) that were precoated with 0.5 mg/ml poly-L-lysine (Sigma Aldrich). Cells were initially plated in Attachment Media (AM; MEM supplemented with 10% horse serum, 33 mM D-glucose, and 1 mM sodium pyruvate) which was replaced with Maintenance Media (MM; Neurobasal [Gibco] supplemented with 33 mM D-glucose, 2 mM GlutaMAX (Invitrogen), 100 units/ml penicillin, 100 µg/ml streptomycin, and 2% B-27 [ThermoFisher]) after 5 hr. Neurons were maintained at 37 C in a 5% $CO_2$ incubator; AraC (5 µM) was added the day after plating to prevent glia cell proliferation. For transfections, neurons (5–8 DIV) were transfected with 0.8–1.2 µg of total plasmid DNA using Lipofectamine 2000 Transfection Reagent (ThermoFisher) and incubated for 18–24 hr; DNA and siRNAs (45 pmol) mixtures were incubated 36–48 hr.

## HeLa culture

HeLa-M (A. Peden, Cambridge Institute for Medical Research) cells were maintained in DMEM (Corning) that was supplemented with 1% GlutaMAX and 10% FBS. Cells were maintained at 37 C in a 5% $CO_2$ incubator. For transfection, cells were plated on uncoated 35 mm glass-bottom dishes (MatTek), transfected with 2 µg of DNA using FuGene 6 (Promega), and incubated for 18 hr. HeLa cells were routinely tested for mycoplasma using a MycoAlert detection kit (Lonza, LT07). Cells were authenticated by STR profiling using GenePrint10 (Promega, B9510) at the DNA Sequencing Facility at The University of Pennsylvania.

## Neuronal mitophagy induction treatments

Mitochondrial damaged was induced via oxidative stress through antioxidant removal or by inhibiting complex III of the electron transport chain using Antimycin A (AA; Sigma Aldrich). For treatments, neuronal media was fully replaced with control MM, AO-free MM (MM where the B-27 supplement has been replaced B-27 supplement, minus antioxidants [ThermoFisher]), or AA MM (MM plus 3 nM AA) for a time course of 1, 2 or 6 hr. For SNAP pulse-chase experiments, neurons were maintained in control media following treatment.

## HeLa mitophagy induction treatments

In HeLa cells, mitochondrial damage was induced by oxidative stress using a bath application of 10 µM AA and 10 µM Oligomycin A. Cells were treated for 90 min then maintained in control media. Recruitment was assessed 0,1, 2, 4, and 6 h after treatment.

## Live-cell imaging

One hour prior to imaging, SNAP (100 nM TMR or JF646; 2 µM Blue 430) and Halo-tag ligands (100 nM) were applied for 30 min, followed by two quick washes and a 30 min washout totaling 1 hr; 2.5 µM Halo-tag ligand was used for HeLa cells. For SNAP pulse-chase experiments, SNAP Block was applied at 1 µM for 2 hr, followed by two quick washes and a 30 min washout. To asses mitochondrial membrane potential, neurons were loaded with 2.5 nM TMRE for 30 min, which occurred during the SNAP or Halo-tag ligand washout. Lysosome acidification was determined using Lysotracker (25–50 nM) that was incubated for 30 min during SNAP or Halo-tag ligand washout. For live-cell imaging, neurons were imaged in Imaging Media (HibernateE [Brain Bits] supplemented with 2% B27 and 33 mM D-glucose). Again, 2% B27, minus antioxidants replaced the standard B27 for AO-free conditions and 3 nM AA was added to the imaging media for AA conditions. HeLa cells were imaged in Leibovitz's (1X) L-15 Medium (Gibco) supplemented with 10% FBS. For TMRE experiments, 2.5 nM TMRE was added to all Imaging Media. Mitochondrial and autophagosome motility was monitored in the mid-axon of 7–8 DIV neurons that were imaged at a rate of 4 timepoints/sec for 5 min. Neurons were imaged in an environmental chamber at 37 C on a Perkin Elmer UltraView Vox spinning disk confocal on a Nikon Eclipse Ti Microscope with a Plan Apochromat Lambda 60 × 1.40 NA and an Apochromat 100 × 1.49 NA oil-immersion objectives and a Hamamatsu EMCCD C9100-50 camera driven by Volocity (Volocity Software, Perkin Elmer). Z-stacks were acquired at 150 nm step-size.

## Intracellular ROS measurements

Hippocampal neurons (20,000–50,000 per well) were plated in 96 well black plate with clear bottom (Corning) that was precoated with 0.5 mg/ml poly-L-lysine, grown for 7 DIV, and treated to induce mitochondrial damage. CellROX (5 µM) was added to each well 30 min prior to the end of treatment (6 h control or AO-free; 2 h AA). Cells were washed three times with PBS and fluorescence was immediately analyzed using a plate reader. For fluorescent imaging of intracellular ROS, CellROX (5 µM) and MitoTracker (20 nM) were added 30 min prior to the end of treatment (6 h control or AO-free; 2 h AA), cells were incubated with CellMask (5 µg/ml) for 5 min, and washed with media three times; neurons were immediately imaged.

## Immunostaining

Neurons were fixed for 10 min using warm 4% PFA (HeLa cells) or 4%/4 %Sucrose (hippocampal neurons). Cells were washed two times with PBS (50 mM NaPO$_4$, 150 mM NaCl, pH 7.4), permeabilized with ice cold methanol for 8 min at −20 C, and blocked for 1 h in blocking solution (5% goat serum and 1% BSA in PBS). Samples were incubated in primary antibodies diluted in blocking solution for 1 h and were washed three times with PBS; these steps was repeated with secondary antibodies. Following the three PBS washes after secondary antibodies, 0.1 mg/ml of Hoechst 33342 reagent (ThermoFisher, H21492) was added to samples and incubated for 10 min; one last PBS wash occurred after this step. Coverslips were mounted in ProLong Gold (Life Technologies) and images were acquired on a Leica DMI6000B inverted epifluorescence microscope with a 63 × 1.4 NA oil-immersion objective and a Hamamatsu ORCA-R2 charge-coupled device camera driven by LAS-AF (Leica Microsystems) or the previously mentioned Perkin Elmer spinning disk confocal.

## Immunoblotting

For fluorescent Western blotting, neurons were washed twice with ice cold PBS and lysed with RIPA buffer (50 mM Tris-HCl pH 7.4, 150 mM NaCl, 0.1% Triton X-100, 0.5% deoxycholate, 0.1% SDS, 1 mM DTT, 1 mM PMSF, and 1x complete protease inhibitor mixture) for 30 min at 4 C. Samples were centrifuged at 4 C for 10 min at 15,800 x g, supernatant was collected, and a BCA assay was performed to determine total protein concentration. Supernatants (20–40 µg) were analyzed by SDS-PAGE and transferred onto PDVF Immobilon FL (Millipore). Membranes were dried for 1 hr, rehydrated in methanol, and stained for total protein (LI-COR REVERT Total Protein Stain). Following imaging of the total protein, membranes were destained, blocked for 1 h in Odyssey Blocking Buffer TBS (LI-COR), and incubated overnight at 4 C primary with antibodies diluted in Blocking Buffer with 0.2% Tween-20. Membranes were washed four times for 5 min in 1xTBS Washing Solution (50 mM Tris-HCl pH 7.4, 274 mM NaCl, 9 mM KCl, 0.1% Tween-20), incubated in secondary antibodies

diluted in Odyssey Blocking Buffer TBS (LI-COR) with 0.2% Tween-20% and 0.01% SDS for 1 hr, and again washed four times for 5 min in the washing solution. Membranes were immediately imaged using an Odyssey CLx Infrared Imaging System (LI-COR).

## Analysis

### Image processing

After live-cell confocal imaging of neurons, z-stack images were processed using Huygens Professional Deconvolution Software (Scientific Volume Imaging, The Netherlands, http://svi.nl) to remove background noise and increase resolution and signal-to noise. It should be noted that fluorescence intensity measurement for TMRE and CellRox experiments were quantified from unprocessed z-stacks. For deconvolution, up to 50 iterations of the Classic Maximum Likelihood Estimation (CMLE) algorithm with theoretical PSF was performed. Background was automatically corrected, the signal to noise ratio was 20–30, and all other settings were default. The number of iterations and signal to noise ratio varied based on the construct. Volume renderings of somal mitochondria and a representative OPTN-positive mitochondrion were generated using the normal shading mode in the volume function of Imaris Software (Bitplane); images were rotated to highlight the mitochondrial network.

### Image analysis

After image processing, mitochondrial fragments and protein ring or puncta formation in z-stack images were manually identified and counted using Fiji (*Schindelin et al., 2012*); only clearly defined mitochondrial localized structures were quantified. Axonal mitochondrial length was quantified using the first frame of each 5 min movie and the Particle Analyzer function in Fiji. The somal mitochondrial aspect ratio was calculated by thresholding single plane images with watershed segmentation and using the Particle Analyzer function. The circularity and area parameters were used to determine the mitochondrial aspect ratio and length, respectively. Cellular acidification of autophagosomes and lysosomes was determined using tandem reporters, where the fluorescence of GFP or SEP quenched in acidic environments. Using single plane images, the total number of single or dual labeled puncta was compared to the total number of puncta for each cell and averaged across trials. The number of $OPTN^{WT}$ and $OPTN^{E478G}$ puncta were manually counted using Volocity. Prism (GraphPad) was used to plot all graphs and determine statistical significance. Adobe Illustrator was used to prepare all figures and images.

### CellROX quantification

For measurements of CellROX using a plate reader, the average CellROX fluorescence intensity from eight wells for each experimental treatment was quantified and normalized to the control. Values for each treatment were averaged across biological replicates. For fluorescence imaging measurements of CellROX, fluorescence intensities were quantified from unprocessed z-stack images. The mean gray value for each soma was determined by averaging the values of five individual areas (2.2 × 2.2 μm square) in the soma. For axons and dendrites, the mean gray value of a minimum of 40 mitochondria were manually measured per image and averaged across biological replicates.

### TMRE quantification

Fluorescence intensities were quantified from max projections of unprocessed z-stack images. In Fiji, the mean gray value for each cell was determined by averaging the values of five individual areas (2.2 × 2.2 μm square) in the soma. In the axon, the mean gray value of mitochondria was manually traced and measured for each cell.

### Mitochondrial content quantification

The somal and mitochondrial volume for each neuron was determined using the volume measurement function in Volocity Quantitation. Mitochondrial content was determined by dividing the mitochondrial volume by the somal volume.

## Immunoblotting quantification

Bands were quantified using Image Studio Software (LI-COR). For each experimental trial, a total protein normalization factor was calculated based on the total protein from each lane. Band intensities for proteins of interest were quantified and divided by the total protein normalization factor corresponding to the same lane. Treatments were normalized relative to the control sample and presented as fold change over control.

## Axonal transport of autophagosomes and mitochondria

In Fiji, kymographs were generated for each axon using a three-pixel line width and the MultipleKymograph plugin. Kymographs were analyzed using MATLAB software (MathWorks). Axonal transport was quantified as described previously (*Maday and Holzbaur, 2016*). The area <100 µm from the axon tip was defined as the distal axon. The mid-axon was the area >100 µm from the axon terminal and from the soma. Directionality of moving organelle was determined by the following: net antero-grade, displacement of $\geq$5 µm within the 5 min imaging window; net retrograde, displacement of $\geq$5 µm within 5 min; and bidirectional/stationary, displacement of <5 µm within 5 min. The percentage of motility was determined for each treatment and averaged across experimental trials. Organelle density was quantified as the number of organelles (from the first frame of the movie) divided by the axonal distance and averaged across biological replicates. The area flux was quantified as the total number of organelles, within the 5 min movie, divided by the length of the axon and time (number of organelles/100 µm/min) and averaged across experimental trials.

### Line scans

In Fiji, line scans were generated using a pixel line width to trace OPTN-positive mitochondria and a three-pixel line width to trace axons and dendrites. Intensity values were normalized by dividing the dataset by its maximum.

## Acknowledgements

The authors would like to thank the members of the Holzbaur laboratory for thoughtful discussion and comments related to this manuscript and Mariko Tokito and Andrea Stavoe for assistance in the design and cloning of constructs. This study was supported by a grant from NIH NINDS (R37 NS060698) to ELFH. CSE was supported by the Howard Hughes Medical Institute Hanna H Gray Fellowship.

## Additional information

### Funding

| Funder | Grant reference number | Author |
| --- | --- | --- |
| National Institute of Neurological Disorders and Stroke | R37 NS060698 | Erika LF Holzbaur |
| Howard Hughes Medical Institute | Hanna H Gray Fellowship | Chantell S Evans |

The funders had no role in study design, data collection and interpretation, or the decision to submit the work for publication.

### Author contributions

Chantell S Evans, Conceptualization, Data curation, Formal analysis, Validation, Visualization, Methodology; Erika LF Holzbaur, Conceptualization, Supervision, Funding acquisition

### Author ORCIDs

Chantell S Evans (iD) https://orcid.org/0000-0001-9401-8604
Erika LF Holzbaur (iD) https://orcid.org/0000-0001-5389-4114

## Ethics

Animal experimentation: This study was performed in accordance with the Guide for the Care and Use of Laboratory Animals of the National Institutes of Health. All animal protocols were approved by the Institutional Animal Care and Use Committee (IACUC) at the University of Pennsylvania (protocol number: 803657). All animals were euthanized prior to harvesting tissue.

## Decision letter and Author response

Decision letter https://doi.org/10.7554/eLife.50260.sa1
Author response https://doi.org/10.7554/eLife.50260.sa2

## Additional files

### Supplementary files

• Supplementary file 1. Key Resources Table. A list of key reagents in this study, including reagent type, designation, source, and identifier (when available or applicable).

• Transparent reporting form

### Data availability

All data generated or analyzed for this study are included in the manuscript and supporting files. Newly generated reagents are available upon request to the authors.

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
