## [Decision Letter]

**Acceptance summary:**

This paper describes a systematic analysis of recruitment of the autophagy cargo adaptor to damaged mitochondria in hippocampal neurons. Under mild oxidative stress, OPTN is recruited to damaged mitochondria and promotes recruitment of additional autophagy proteins and formation of an autophagosome, occurring robustly in the cell soma. OPTN- and LAMP1-positive structures remained in a non-acidic state for extended time periods, suggesting that acidification of lysosomes containing mitochondria is rate-limiting for degradation. This work is relevant to Parkinson's disease and ALS.

**Decision letter after peer review:**

Thank you for submitting your article "Mitophagosome acidification is rate-limiting in the maintenance of the somal mitochondrial network in neurons" for consideration by *eLife*. Your article has been reviewed by three peer reviewers, and the evaluation has been overseen by a Reviewing Editor and Suzanne Pfeffer as the Senior Editor. The reviewers have opted to remain anonymous.

The reviewers have discussed the reviews with one another and the Reviewing Editor has drafted this decision to help you prepare a revised submission.

This paper addresses the interesting question of how the autophagy cargo adaptor OPTN is employed in post-mitotic neurons to remove damaged mitochondria downstream of the Parkin ubiquitin ligase. Parkin is recruited to damaged mitochondria in response to PINK1 activation and catalyzes ubiquitylation of a number of proteins on the outer membrane of mitochondria. OPTN then binds to ubiquitin chains to promote recruitment of the autophagy machinery. The authors characterize this process in neurons in the presence of "endogenous" mitochondrial damage. Through a series of studies, they provide evidence that mitoautophagosomes acidify slowly, suggesting that acidification is a rate-limiting step in mitophagy.

While the reviewers all feel that the paper is timely and interesting, there are several concerns that need to be addressed before the paper can be considered further. These concerns generally fall into 4 areas:

1) OPTN overexpression. The paper extensively employs ectopic expression of tagged OPTN. Although this is very useful for detection of OPTN-coated mitochondria, it raises issues in the context of the model wherein mitolysosome acidification is rate-limiting in neurons. Depending upon the levels of overexpression, it is possible that autophagosome formation is accelerated in some way that makes acidification seem slow (rate-limiting). The paper also tries to make comparisons with HeLa cells in this context. Because the abundance of endogenous OPTN relative to the tagged OPTN isn't shown in either the case of neurons or HeLa cells, it is difficult to know the extent to which this may be an issue in the experiments that are shown. It is important to show the levels of overexpression either in the population, or preferably in individual cells using endogenous OPTN antibodies. This issue is also important in light of the fact that there are a substantial number of OPTN puncta that are not co-localized with mitochondria. Are these aggregates resulting from overexpression or are they something else? DO such OPTN puncta appear in neurons using endogenous antibodies? In terms of overexpression of OPTN accelerating early steps, it may be possible to visualize mitoautophagosomes using GFP-LC3, and since this is functioning at a step downstream of OPTN recruitment, it may allow an independent assessment of whether acidification is rate-limiting.

2) Lysosome acidification as the rate-limiting step. All three reviewers had concerns about this strong conclusion, which is also featured in the Title of the paper. This reflects, in part, the OPTN overexpression as discussed above, as well as the approaches being used to examine acidification. The reviewer' feel that a more direct method, including measurement of mitophagic flux using Keima or mito-QC would strengthen the results. Is there evidence of mitochondria inside lysosomes that have not yet either activated the Keima red-shift or converted the yellow GFP-mCherry signal to mCherry only, which would indicate non-instantaneous acidification, and if so, what is the time constant for conversion to the acidic state? There is also some concern that the LAMP1 marker used for marking lysosomes may also, when overexpressed, mark autophagosomes, making the designation of mitolysosomes complicated. It may be possible to follow local diffusion (i.e. signal expansion) of lysotracker from a lysosome into an OPTN-positive autophagosome in order to measure the length of time from full OPTN recruitment to acidification. Previous studies have used this approach to examine the kinetics by live-cell imaging.

3) A third issue brought up by the reviewers concerns the question of whether the pathway being examined is the Parkin-PINK1 pathway or not. This is the expectation given the fact that OPTN is recruited to the damaged mitochondria, but isn't formally demonstrated. Might it be possible to deplete PINK1 or Parkin by shRNA and examine the number of OPTN-positive mitochondria that appear over time in individual cells relative to cells transfected with control shRNA?

4) The current work implies that the delay in acidification is specific to mitophagosomes. However, the acidification rate of autophagosomes in general, under the authors experimental conditions, was not tested. The authors could monitor the formation of OPTN -ve LC3 puncta using their mild depolarising media and determine if these equally take as long to become acidified.

Reviewer #1:

This paper describes an analysis of mitophagy in neuronal cells. Mitophagy is perhaps best understood in the context of PINK1 and PARK2-dependent mitochondrial ubiquitylation and targeting to the autophagosome via an OPTN-TBK1-dependent pathway (although NDP52 and TAX1BP1 can also function in this pathway redundantly with OPTN). The majority of studies examining PARK2-dependent mitophagy have been done in cancer cell lines overexpressing PARK2 and there has been conflicting data for PARK2-dependent mitophagy in neuronal cell lineages. This paper tries to address some of these conflicting data and also presents a new model related to lysosomal acidification as being rate-limiting for mitophagy in neurons.

In mito-QC or mito-Keima mice that lack PINK1 or PARK2, there is no discernable defect in mitophagic flux across all/most tissues, indicating that the majority of mitophagic flux is through a PARK2-independent pathway. However, this doesn't necessarily mean that there isn't a small amount of flux (not detectable by the methods used) that is still PARK2-dependent in specific classes of neurons. Additionally, previous studies have used various types of mitochondrial depolarizing agents to examine mitophagy in neurons in culture. In some experiments, mitophagy is initiated in the axon while in others, it occurs in the soma. While there are discrepancies in some of the conclusions, it's also probably fair to say that some of the conclusions are not really based on strong experiments. As such, there is still much that is unclear concerning when, where, and how mitochondria are degraded by autophagy in neurons.

This study uses removal of anti-oxidants (AOF) from the media as a way to induce endogenous mitochondrial damage, with the idea that it will be possible to examine mitophagy mechanisms in a more physiological setting. The paper is primarily based on imaging.

Major comments:

Throughout the paper, the authors employ overexpression of Halo-OPTN. There is the distinct possibility that overexpression of OPTN can artificially accelerate steps in mitophagy that are downstream of PARK2-action on mitochondria. This complicates the interpretation of the data in terms of the rates of mitochondrial turnover and the conclusions related to lysosomal acidification being limiting. The recent finding that autophagy receptors (NDP52 in particular) can recruit ULK1-FIP200 raises the question as to whether having additional receptor in the cell could accelerate autophagosome formation around ubiquitylated mitochondira. Additionally, in Figure 1I, there appear to be many Halo-OPTN puncta that are not co-localizing with mitochondria. It is unclear what these puncta are. Do they reflect Halo-OPTN aggregates, for example, resulting from overexpression? Or are they OPTN localized on other autophagic cargo?

In Figure 2 the authors examine localization of PARK2, TBK1 and OPTN with mitochondria and correlate it with TMRE signal. In some sense, it's a bit hard to judge because they are only showing AOF treated samples and so there isn't a systematic quantification of the number of OPTN-coated mitochondria in the absence of oxidative stress. There are a lot of OPTN-positive, TBK1-positive puncta whose identity are unknown. I also am unsure what the authors mean concerning Figure 2G, where the authors say that the number of OPTN on linear mitochondria are "low and similar in either conditions". The value is ~29%, which is much higher that the OPTN on fragmented mitochondria, so this doesn't seem to make sense. I am also not sure about the argument made concerning the aspect ratio with and without AOF in Figure 2H. The highest signal in the control sample is also the <1.5 ratio, and this only marginally increased with AOF. So, it would seem to be hard to conclude that this small mitochondria directly represents the ones that are subsequently tagged with OPTN.

Figure 3 looks at the frequency of OPTN encircled mitochondria in axons, dendrites and soma, indicating that virtually no OPTN-positive mitochondria are seen in the axon. The frequency of OPTN-positive mitochondria relative to the total mitochondrial volume in the soma indicates that with AOF, there is a very small amount of damaged mitochondria relative to the total mitochondrial volume. Given this very low value, it would appear to take many axons in order to achieve a similar mitochondrial volume in order to directly compare the frequency of damage and capture by OPTN. So based on a stochastic damage argument, it may be very difficult to detect damaged mito in the axon but that doesn't necessarily mean it doesn't happen there. So, the conclusion in subsection “OPTN-mediated mitophagy occurs preferentially in the somatodendritic compartment of hippocampal neurons” might be too strong.

The experiment in Figure 5 is a bit hard to interpret. Since LAMP1 is overexpressed, it is possible that it is marking organelles such as endosomes, in addition to lysosomes. In any event, the apparent number of LysoT positive puncta is far lower than the number of LAMP1 positive puncta, albeit not in the same cells. The major conclusion is that lysosomes are acidified more slowly in neurons, and results in HeLa cells overexpressing OPTN and LAMP1 are used to compare the rates of lysosome acidification. Interpretation of this experiment is difficult because there is no indication of the levels of expression of OPTN/cell. If OPTN overexpression can accelerate the assembly of a fully coated autophagosome around the damaged mitochondria, and therefore the rate of fusion with a lysosome since full closure is required, then the difference in the apparent rate of acidification could simply represent the levels of OPTN in each cell being examined. Since there are no measurements of OPTN levels/per cell and specifically in the cells that are being imaged, it makes it challenging to make the conclusion that rates of acidification are intrinsically slower in neurons. If much less overexpressed OPTN is present in neurons than HeLa, this could explain the difference seen. The authors examined cathepsin cleavage, and find only minor differences as measured in a western blot assay. Since cathepsin processing also requires low lysosomal pH, this a priori would suggest no issue with lysosomal acidification at a global level in the neurons, raising the question of precisely how acidification is regulated.

With regards to the conclusions in Figure 5, an ultrastructure analysis in the cited Cheng et al., 2018 paper suggests that LAMP1 can be present on immature double membrane autophagosome-like structures. Is it possible that some of the LAMP1-positive signals lacking LysoT positivity are actually autophagosomes that are LAMP1 positive but not yet fused with a lysosome? Could LAMP1 overexpression affect the interpretation of the experiments?

Subsection “Low-level induction of ROS leads to selective mitochondrial damage without compromising the overall mitochondrial network”. In the two color old/new mitochondrial experiment, it's hard to imagine how the age can be precisely controlled using the described protocol. The SNAP tag for mito localization contains a COXVIII MTS sequence. As such, it would seem like any mitochondria with a functional translocon could import the SNAP-tag protein, regardless of whether it is old or new, however the authors state that the two populations are "distinct". The interpretation that old mitochondria can import the newly synthesized snap tag is consistent with most of the mitochondria being double labeled (Figure 6C). Because the mass of old mitochondria is much larger than the mass of new mitochondria that has been made since the new dye was added (dye added for 1 hour) and the number of those mitochondria not undergoing intermixing by fusion is small, the frequency with which such a "new" mitochondria would be identified as damaged and associated with OPTN would be particularly low, relative to the much more abundant old mito. So it seems possible, that this experiment doesn't really have the statistical power to rule out similar rates of old and new mitochondrial engulfment. The authors argue that the absence in green signal means that they are exclusively old, but it also may be that the signal to noise isn't equivalent. The labeling period for old was 24 hours while the labeling for new was only 1 hour. Therefore, all things being equal in terms of mitochondrial assembly/fission/fusion, there should be far less signal for new snap relative to old snap/per individual mito.

In Figure 7, the authors examine an OPTN mutant found in ALS, and see general but small effects on mitochondrial size. This experiment is somewhat limited by the fact that OPTN deletion by siRNA is only ~60%, and it is possible that the overexpressed OPTN mutant (not sure how much it is overexpressed as they didn't show a western of the transfected cells) may also act as a dominant negative – binding TBK1 but not Ub chains. What isn't clear is whether the authors have missed an opportunity to statistically determine whether the OPTN mutant encircles damaged mito. The experiments are largely limited to mito morphology.

An important weakness of the analysis is the absence of a direct demonstration of mitophagic flux.

Reviewer #2:

In their manuscript, Evans and Holzbaur present data arguing two main points regarding mitophagy in neurons: (1) that mitophagy occurs mainly in the soma and not in axons, contrary to some previous work by others, and (2) that neuronal mitophagy is significantly slower relative to other cell types, as result of slow acidification of mitoautophagosomes. Although the experiments are generally well-performed additional evidence is required in support of their main conclusions.

Major comments:

1) The implication throughout the paper is that the optineurin rings forming around mitochondria in response to mild oxidative stress are due to PINK1/Parkin dependent mitophagy. This should be demonstrated with knockdown or knockout of PINK1 and Parkin. This is particularly important given lack of clarity on the role of PINK1/Parkin mitophagy in neurons in response to more physiologically relevant stressors.

2) A major finding of the manuscript is that mitophagy occurs primarily in the cell soma and not the axons, in contrast to the findings of Ashrafi et al. As a rationale for this discrepancy, they suggest that they are applying a milder stress (withholding antioxidants from neuronal media) that does not lead to global bioenergetic collapse. This explanation doesn't seem satisfactory, however, as Ashrafi et al., also, used a variety of experimental setups to induce local damage mitochondria without affecting the overall bioenergetics of the neuron. An alternative explanation for the discrepancy might be that the particular exposure used by Evans and Holzbaur (withholding antioxidants) preferentially increases oxidative stress and mitochondrial damage in the cell soma as opposed to the axons. Does ROS increase equally in the axon and cell body with antioxidant withdrawal? Does the authors treatment result in TMRE negative mitochondria in axons that are not captured by Optineurin in contrast to the cell soma? When the authors treat with antimycin for 2 hours (which increases ROS locally at the mitochondria) do they see the same preferential increase in mitophagy in the soma or are mitochondria in the axons also targeted? Ashrafi saw axonal events as soon as 45 minutes, do the authors see axonal events at these early timepoints? Ashafri noted increased axonal events with the use of lysosomal inhibitors. If these are used by the authors, do they then see axonal events?

3) In Figure 5 the authors compare the speed of mitophagosome maturation in neurons and HeLa cells. They conclude that the difference in speed reflects a difference in processes downstream of mitochondrial ubiquitination, but the level mitochondrial ubiquitination is likely very different in neurons with endogenous Parkin subjected to mild oxidative stress and HeLa cells over-expressing Parkin and fully depolarized with OA. Could this account for the difference in maturation rate? Do the authors still see a difference in the speed of mitophagosome maturation when the two cell types are subjected to more equal treatment – e.g., in neurons and HeLa cells that are both overexpressing Parkin and are subjected to the same oxidative stress (e.g., antimycin)? Additionally, the authors note that acidification of mitoauthophagosomes is delayed in neurons? How long is it delayed? What do they see at time points longer than one day?

Reviewer #3:

In the manuscript by Evans and Holzbaur, the authors examine the dynamics of OPTN recruitment to mitochondria and subsequent delivery to lysosomes following mild oxidative stress in primary rat hippocampal neurons. The authors mainly use transfected neurons and live-cell microscopy and find that OPTN is rapidly recruited to a subset of mitochondria, primarily in the soma, following anti-oxidant removal from the cell media. This then correlated with subsequent recruitment of mitophagy and lysosomal markers. Surprisingly, and the main take home message from the manuscript, was that the OPTN puncta on mitochondria do not become efficiently acidified (as visualized by col-localisation with lysotracker), from which the authors conclude that this is a rate-limiting step during neuronal mitophagy.

This is an intriguing manuscript with many interesting observations. The identification of "mild" depolarising conditions that trigger a mitophagy response will be useful to the field and certainly does appear more physiological that the use of CCCP or oligomycin/antimycin. The microscopy images are impressive and convincing in what they show. However, I think more work is needed to determine whether this phenomenon is specific to mitophagosomes and neurons in general.

Major comments:

1) I think more robust analyses of lysosome acidification are needed. It is surprising and very interesting that there is a large delay in acidification of mitolysosomes. Is this a general phenomenon, i.e. are acidified lysosomes scarce in these cells? What is the percentage of LAMP1 structures (regardless of co-localising with OPTN) that are LysoTracker +ve? If it is low, could this mean that most of the LAMP1-GFP is not in lysosomes? Could the authors use another lysosomal marker to confirm?

2) Related to the above point, the authors only use LysoTracker to estimate lysosomal acidification. Is there another way the authors could estimate this – for example using ratiometric fluorophores, such as Keima or mCherry-GFP?

3) Is the delay in acidification specific to mitolysosomes? What about autophagosomes in general? The authors could monitor the formation of OPTN -ve LC3 puncta under their conditions and determine if these equally take as long to become acidified. The authors may wish to discuss Ralph Nixon's work (e.g. Boland et al., 2008) that argues autophagosome fusion with lysosomes is very efficient in neurons.

4) Are the authors sure it is mitophagosome acidification that is rate-limiting and not an earlier step in the pathway? For example, is it possible that the autophagosome is not fully sealed (despite appearing circular by fluorescence microscopy). The authors could look at early autophagosome markers such as ULK1, ATG5 or WIPI that are not present on mature autophagosomes. Or mature autophagosome markers such as STX17.

5) The authors imply that this delay is specific to neurons, yet they only look in hippocampal neurons and compare this to HeLa cells overexpressing Parkin. I think the authors need to provide some more data in relevant cell lines for a better comparison. What about cortical neurons, as well and comparing with primary fibroblasts (which should contain all the relevant Parkin pathway components)?

[Editors' note: further revisions were requested prior to acceptance, as described below.]

Thank you for resubmitting your work entitled "Degradation of engulfed mitochondria is rate-limiting in Optineurin-mediated mitophagy in neurons" for further consideration by *eLife*. Your revised article has been evaluated by Suzanne Pfeffer (Senior Editor) and a Reviewing Editor.

The manuscript has been improved but there are some remaining issues that need to be addressed before acceptance, as outlined below:

Based on the comments of reviewer 3, a couple of changes in the manuscript are still needed. First, some data that are described in the rebuttal don't appear in the paper, so Figure 4 may need to be updated to include data that were discussed in the rebuttal. In addition, there are a couple of suggested changes to the text that will need to be addressed.

Reviewer #1 and 2:

Overall, the authors have done a good job of addressing the previous comments. I have no further concerns and feel that the paper should be accepted.

Reviewer #3:

This is a re-review of the manuscript by Evans et al., describing the slow maturation/impaired acidification of mitophagosomes in the soma of neurons. I think the authors have done a great job in strengthening their manuscript – my main concern was with the lysosomal acidification studies and the use of the dual reporters means the data are much more convincing. I just have a few small points remaining:

1) The LAMP2 data, mentioned in subsection” Acidification of OPTN-positive mitochondria is a rate-limiting step in neuronal mitophagy” and in the comments to reviewers does not seem to be in the manuscript. The authors mention Figure 4E, but these are only LC3-related. Please include the data.

2) Likewise, the authors show data in the reviewers comments that the OPTN/LC3 structures are WIPI -ve. I do not see these data in the manuscript. This shows that the structures are not early autophagosomal in nature and thus they will be of interest to readers (and not just this reviewer) and so should be included (at least in the supplemental). I'm assuming WIPI2B is expressed in these cells and does indeed form puncta with LC3 when autophagy is induced?

3) I also take on the point that the authors have made in response to my concern raised in point 5 – i.e. that HeLa cells overexpressing Parkin are a suitable system to compare to the neurons. The HeLa cell has been instrumental in the field in terms of building up the mechanism of PINK/Parkin mitophagy. But this is not a "normal" cell. Normally, it does not undergo this pathway as it does not express Parkin! I think to avoid doing more experiments, the authors can be more accurate in what they state and say that this delay in acidification does not occur in HeLa cells expressing Parkin. For example in subsection “Acidification of OPTN-positive mitochondria is a rate-limiting step in neuronal mitophagy”: "…..much slower than the corresponding time course in non-neuronal cells." Should be replaced with "…..much slower than the corresponding time course in HeLa cells."

---

## [Author Response]

This paper addresses the interesting question of how the autophagy cargo adaptor OPTN is employed in post-mitotic neurons to remove damaged mitochondria downstream of the Parkin ubiquitin ligase. Parkin is recruited to damaged mitochondria in response to PINK1 activation and catalyzes ubiquitylation of a number of proteins on the outer membrane of mitochondria. OPTN then binds to ubiquitin chains to promote recruitment of the autophagy machinery. The authors characterize this process in neurons in the presence of "endogenous" mitochondrial damage. Through a series of studies, they provide evidence that mitoautophagosomes acidify slowly, suggesting that acidification is a rate-limiting step in mitophagy.While the reviewers all feel that the paper is timely and interesting, there are several concerns that need to be addressed before the paper can be considered further. These concerns generally fall into 4 areas:1) OPTN overexpression. The paper extensively employs ectopic expression of tagged OPTN. Although this is very useful for detection of OPTN-coated mitochondria, it raises issues in the context of the model wherein mitolysosome acidification is rate-limiting in neurons. Depending upon the levels of overexpression, it is possible that autophagosome formation is accelerated in some way that makes acidification seem slow (rate-limiting). The paper also tries to make comparisons with HeLa cells in this context. Because the abundance of endogenous OPTN relative to the tagged OPTN isn't shown in either the case of neurons or HeLa cells, it is difficult to know the extent to which this may be an issue in the experiments that are shown. It is important to show the levels of overexpression either in the population, or preferably in individual cells using endogenous OPTN antibodies. This issue is also important in light of the fact that there are a substantial number of OPTN puncta that are not co-localized with mitochondria. Are these aggregates resulting from overexpression or are they something else? DO such OPTN puncta appear in neurons using endogenous antibodies? In terms of overexpression of OPTN accelerating early steps, it may be possible to visualize mitoautophagosomes using GFP-LC3, and since this is functioning at a step downstream of OPTN recruitment, it may allow an independent assessment of whether acidification is rate-limiting.

We thank the reviewers for their comments and agree that it is important to consider the expression level of OPTN when discussing the rates of neuronal mitophagy. As requested, we compared OPTN expression in non-transfected and Halo-OPTN transfected cells (representative images are provided in Author response image 1). Using an OPTN antibody and a Halo ligand, we observed both cytosolic localization and the presence of discrete OPTN puncta in neurons and HeLa cells expressing endogenous OPTN and in cells expressing labeled OPTN.

**Author response image 1. respfig1:** Representative images of hippocampal neurons (**A**) and HeLa cells (**B**) comparing OPTN expression levels in non-transfected and Halo-OPTN transfected cells. aOPTN polyclonal antibody; Abcam ab23666. Scale bar, 100 μm.

Representative images were previously included in Figure 2—figure supplement 1 and a representative western blot is now included in Figure 8—figure supplement 1. These observations are consistent with previous work demonstrating the punctate appearance of endogenous OPTN (Park et al., 2006). We also confirmed these findings using two independent aOPTN antibodies (Abcam ab23666 and Cayman Chemical 100000). To further avoid over-expression artifacts, we select neurons within a narrow fluorescence range in our imaging experiments.

We determined that ~60-70% of OPTN puncta are not associated with mitochondria (now included in Figure 1—figure supplement 1D). OPTN has been implicated in other processes, such as aggrephagy (Korac et al., 2013). As a result, these Halo-OPTN puncta may be localized to other autophagic cargos. Consistent with this, we found that roughly 50% of nonmitochondrial associated OPTN puncta colocalized with LAMP1-EGFP, as shown in Author response image 2:

**Author response image 2. respfig2:** Representative images of a hippocampal neuron showing Halo-OPTN puncta are found in LAMP1-EGFP containing organelles in the soma. These puncta are negative for mitochondria. Examples of events are indicated by the red arrows. Scale bar, 5 μm.

Finally, we transfected neurons with a mitochondrial marker (Mito-SNAP) and a tandem mCherry-GFP-LC3 to look at acidification in the absence of overexpressed OPTN. We continued to find consistent evidence of spherical mitochondria that colocalized with both GFP and mCherry, further corroborating our findings that while these mitochondria are engulfed within autophagosomes, they are not yet acidified (as determined by the unquenched nature of the GFP signal on the tandem fluorescent LC3 reporter). A representative image is provided in Author response image 3.

**Author response image 3. respfig3:** Representative images of a hippocampal neuron following antioxidant removal for 6 hours. An example of a LC3-positive mitochondrion in the absence of overexpressed OPTN is shown. Both the GFP and mCherry of the tandem LC3 marker are present, illustrating the damaged mitochondrion is sequestered in a nonacidified organelle. Scale bar, 1 μm.

2) Lysosome acidification as the rate-limiting step. All three reviewers had concerns about this strong conclusion, which is also featured in the Title of the paper. This reflects, in part, the OPTN overexpression as discussed above, as well as the approaches being used to examine acidification. The reviewer' feel that a more direct method, including measurement of mitophagic flux using Keima or mito-QC would strengthen the results. Is there evidence of mitochondria inside lysosomes that have not yet either activated the Keima red-shift or converted the yellow GFP-mCherry signal to mCherry only, which would indicate non-instantaneous acidification, and if so, what is the time constant for conversion to the acidic state? There is also some concern that the LAMP1 marker used for marking lysosomes may also, when overexpressed, mark autophagosomes, making the designation of mitolysosomes complicated. It may be possible to follow local diffusion (i.e. signal expansion) of lysotracker from a lysosome into an OPTN-positive autophagosome in order to measure the length of time from full OPTN recruitment to acidification. Previous studies have used this approach to examine the kinetics by live-cell imaging.

We agree that additional methods were needed to further strengthen our conclusion that lysosome acidification is rate-limiting in neuronal mitophagy. To address this point, we utilized the mitochondrial matrix directed tandem fluorescent reporter Cox8-EGFP-mCherry to look at mitophagic flux (Rojansky et al., 2016); this construct is similar to the *mito*-QC previously described (Allen et al., 2013; McWilliams et al., 2016). We identified OPTN-positive mitochondria and looked for the presence or absence of EGFP fluorescence coincident with mCherry fluorescence to identify non-acidified and acidified organelles. Using this assay, we found OPTN-positive mitochondria that were dual labeled with the tandem reporter, suggesting they were in non-acidified compartments. This data is now included in the revised manuscript (please see Figure 6).

To use LAMP1 as a second, independent reporter for acidified degradation-competent organelles, we utilized a dual labeled SEP-LAMP1-RFP construct where superecliptic pHluorin (SEP) is fused to the lumen of LAMP1-RFP (Farias et al., 2017). The fluorescent signal of the SEP will quench in acidic environments (pH < 6; (Sankaranarayanan et al., 2000)). We felt that a dual reporter system would provide more reliable measurements than the local diffusion of lysotracker from a lysosome. As a control experiment, we used this construct to determine the number of acidified lysosomes per cell (see Figure 5—figure supplement 1) and found it to be consistent with other methods used. We then identified OPTN-positive mitochondria and quantified the relative percentage that were dual or single labeled with the tandem LAMP1 probe. We determined the majority of sequestered mitochondria were dual positive for green and red fluorescence with the SEP-LAMP1-RFP tandem probe, again suggesting that this represents a substantial population of non-acidified organelles (see Figure 6 and Figure 6—figure supplement 1).

3) A third issue brought up by the reviewers concerns the question of whether the pathway being examined is the Parkin-PINK1 pathway or not. This is the expectation given the fact that OPTN is recruited to the damaged mitochondria, but isn't formally demonstrated. Might it be possible to deplete PINK1 or Parkin by shRNA and examine the number of OPTN-positive mitochondria that appear over time in individual cells relative to cells transfected with control shRNA?

To formally demonstrate whether this pathway is PINK1/Parkin-dependent, we performed Parkin knock-down and rescue experiments. The percent of neurons with OPTN-positive mitochondria was determined in neurons following antioxidant removal for 1 hour, see Figure 2—figure supplement 2 of the revised manuscript. We observed an overall 30% reduction in Parkin expression by immunoblot analysis of cell lysates, but it is important to note that this measurement is an underestimate of the reduction in Parkin expression in the imaged neurons, as transfection rates are low in hippocampal neurons and the immunoblot measurement is from an entire plate of cells and so includes neurons that did not take up siRNA. When imaging neurons, we selected cells that were transfected and therefore have also taken up siRNA. Importantly, this level of knock-down was sufficient to significantly inhibit the number of cells with OPTN-positive rings, from ~35% down to less than 10%. This deficit was rescued by reexpression of WT Parkin. These data along with our recruitment data indicate that the PINK1-Parkin pathway is involved, although we cannot rule out the involvement of other possible ROS-induced E3 ligases in addition to Parkin. This point is directly addressed in the Discussion section.

4) The current work implies that the delay in acidification is specific to mitophagosomes. However, the acidification rate of autophagosomes in general, under the authors experimental conditions, was not tested. The authors could monitor the formation of OPTN -ve LC3 puncta using their mild depolarising media and determine if these equally take as long to become acidified.

We agree that it is important to determine whether our mitophagy induction paradigm alters autophagosome acidification in general or if it is specific to mitophagosomes. As requested, we identified OPTN-negative LC3 puncta and monitored acidification using the tandem mCherry-GFP-LC3 reporter. We determined that 75% of autophagosomes in a cell were acidified and that there was no significant difference in acidification between control or AO-free treated conditions at 1 or 6 hours. Additionally, we monitored lysosome acidification and observed similar results. See Figure 5—figure supplement 1 of the revised manuscript.

Reviewer #1:This paper describes an analysis of mitophagy in neuronal cells. Mitophagy is perhaps best understood in the context of PINK1 and PARK2-dependent mitochondrial ubiquitylation and targeting to the autophagosome via an OPTN-TBK1-dependent pathway (although NDP52 and TAX1BP1 can also function in this pathway redundantly with OPTN). The majority of studies examining PARK2-dependent mitophagy have been done in cancer cell lines overexpressing PARK2 and there has been conflicting data for PARK2-dependent mitophagy in neuronal cell lineages. This paper tries to address some of these conflicting data and also presents a new model related to lysosomal acidification as being rate-limiting for mitophagy in neurons.In mito-QC or mito-Keima mice that lack PINK1 or PARK2, there is no discernable defect in mitophagic flux across all/most tissues, indicating that the majority of mitophagic flux is through a PARK2-independent pathway. However, this doesn't necessarily mean that there isn't a small amount of flux (not detectable by the methods used) that is still PARK2-dependent in specific classes of neurons. Additionally, previous studies have used various types of mitochondrial depolarizing agents to examine mitophagy in neurons in culture. In some experiments, mitophagy is initiated in the axon while in others, it occurs in the soma. While there are discrepancies in some of the conclusions, it's also probably fair to say that some of the conclusions are not really based on strong experiments. As such, there is still much that is unclear concerning when, where, and how mitochondria are degraded by autophagy in neurons.This study uses removal of anti-oxidants (AOF) from the media as a way to induce endogenous mitochondrial damage, with the idea that it will be possible to examine mitophagy mechanisms in a more physiological setting. The paper is primarily based on imaging.Major comments:1) Throughout the paper, the authors employ overexpression of Halo-OPTN. There is the distinct possibility that overexpression of OPTN can artificially accelerate steps in mitophagy that are downstream of PARK2-action on mitochondria. This complicates the interpretation of the data in terms of the rates of mitochondrial turnover and the conclusions related to lysosomal acidification being limiting. The recent finding that autophagy receptors (NDP52 in particular) can recruit ULK1-FIP200 raises the question as to whether having additional receptor in the cell could accelerate autophagosome formation around ubiquitylated mitochondira. Additionally, in Figure 1I, there appear to be many Halo-OPTN puncta that are not co-localizing with mitochondria. It is unclear what these puncta are. Do they reflect Halo-OPTN aggregates, for example, resulting from overexpression? Or are they OPTN localized on other autophagic cargo?

Halo-OPTN does form puncta that do not colocalize with mitochondria. However, we observed a similar distribution of endogenous OPTN in the soma of neurons and in HeLa cells, please see Figure 2—figure supplement 1 and additional images above. These observations are consistent with previous work demonstrating the punctate appearance of endogenous OPTN (Park et al., 2006).

It is possible that some of these puncta localize with autophagic cargos. OPTN has been suggested to function in aggrephagy and some of these Halo-OPTN puncta could localize with aggregated proteins (Korac et al., 2013). Consistent with this, we observed that many of the Halo-OPTN puncta colocalize with EGFP-LC3 and LAMP1-EGFP (see above for representative images). However, it is difficult to determine whether these puncta represent OPTN-mediated clearing events or overexpressed protein that is being cleared from the cell, as both are destined for autophagosome engulfment and degradation via lysosomal fusion.

2) In Figure 2 the authors examine localization of PARK2, TBK1 and OPTN with mitochondria and correlate it with TMRE signal. In some sense, it's a bit hard to judge because they are only showing AOF treated samples and so there isn't a systematic quantification of the number of OPTN-coated mitochondria in the absence of oxidative stress. There are a lot of OPTN-positive, TBK1-positive puncta whose identity are unknown. I also am unsure what the authors mean concerning Figure 2G, where the authors say that the number of OPTN on linear mitochondria are "low and similar in either conditions". The value is ~29%, which is much higher that the OPTN on fragmented mitochondria, so this doesn't seem to make sense. I am also not sure about the argument made concerning the aspect ratio with and without AOF in Figure 2H. The highest signal in the control sample is also the <1.5 ratio, and this only marginally increased with AOF. So, it would seem to be hard to conclude that this small mitochondria directly represents the ones that are subsequently tagged with OPTN.

On average ~95% of OPTN-positive mitochondria were TMRE negative in both control and AO-free conditions. Since mitophagy can occur at a basal level, OPTN-positive mitochondria were found in both control and treated conditions, but the number of events was higher in AO-free treated conditions compared to control (see Figure 3D, F for the percent of cells with OPTN-positive mitochondria). In Figure 2D-F, we observed two types of localization that occurred using upstream mitophagy-associated proteins, the formation of rings around damaged mitochondria and puncta that colocalized with isolated mitochondrial fragments. We highlighted examples of each using arrows, but every event was not highlighted.

As shown in Figure 1—figure supplement 1D, ~30-40% of total OPTN puncta were found on mitochondria. Of this population, most were localized to linear mitochondria under both control and AO-free conditions. Although, the percent associated with rounded mitochondria was observed to increase significantly upon mild stress in the AO-free condition. We have now clarified this description in the revised manuscript.

We measured the mitochondrial aspect ratio to determine whether we could quantify minor changes in the mitochondrial network. Overall, we observed a small but significant decrease in the mitochondrial aspect ratio. The binned data illustrate that the fraction of mitochondria with an aspect ratio of ≤1.5 was greater in neurons treated with AO-free media compared to control (now Figure 1—figure supplement 1A). Additionally, we measured the diameter of OPTN-positive mitochondrial rings and found they were on average ~1.2 µm (data not shown). Together these data suggest that some of the small mitochondria represent the ones that are engulfed by OPTN.

Figure 3 looks at the frequency of OPTN encircled mitochondria in axons, dendrites and soma, indicating that virtually no OPTN-positive mitochondria are seen in the axon. The frequency of OPTN-positive mitochondria relative to the total mitochondrial volume in the soma indicates that with AOF, there is a very small amount of damaged mitochondria relative to the total mitochondrial volume. Given this very low value, it would appear to take many axons in order to achieve a similar mitochondrial volume in order to directly compare the frequency of damage and capture by OPTN. So based on a stochastic damage argument, it may be very difficult to detect damaged mito in the axon but that doesn't necessarily mean it doesn't happen there. So, the conclusion in subsection “OPTN-mediated mitophagy occurs preferentially in the somatodendritic compartment of hippocampal neurons” might be too strong.

We thank the reviewer for the insightful comment and agree that it may be difficult to detect damaged mitochondria in the axon based on the stochastic damage argument. We determined the somal mitochondrial content was ~0.2 µm^3^/µm^3^ (Figure 1F) and the axonal mitochondrial content to be ~0.02 µm^3^/µm^3^ (data not shown). Based on these values and the number of somas analyzed in Figure 3, 210-280 axons would need to be analyzed. Since this is not the ideal system for such analysis, we cannot definitively rule out the possibility that an event occurred that we did not image. As a result, we have changed the statement to say “with limited events in dendrites and axons.” Additionally, we discuss this caveat and provide more axonal analysis in the Results section.

The experiment in Figure 5 is a bit hard to interpret. Since LAMP1 is overexpressed, it is possible that it is marking organelles such as endosomes, in addition to lysosomes. In any event, the apparent number of LysoT positive puncta is far lower than the number of LAMP1 positive puncta, albeit not in the same cells. The major conclusion is that lysosomes are acidified more slowly in neurons, and results in HeLa cells overexpressing OPTN and LAMP1 are used to compare the rates of lysosome acidification. Interpretation of this experiment is difficult because there is no indication of the levels of expression of OPTN/cell. If OPTN overexpression can accelerate the assembly of a fully coated autophagosome around the damaged mitochondria, and therefore the rate of fusion with a lysosome since full closure is required, then the difference in the apparent rate of acidification could simply represent the levels of OPTN in each cell being examined. Since there are no measurements of OPTN levels/per cell and specifically in the cells that are being imaged, it makes it challenging to make the conclusion that rates of acidification are intrinsically slower in neurons. If much less overexpressed OPTN is present in neurons than HeLa, this could explain the difference seen. The authors examined cathepsin cleavage, and find only minor differences as measured in a western blot assay. Since cathepsin processing also requires low lysosomal pH, this a priori would suggest no issue with lysosomal acidification at a global level in the neurons, raising the question of precisely how acidification is regulated.

We quantified the number of LAMP-positive organelles that are LysoT positive in cells overexpressing LAMP1 or LAMP2, and determined that ~89% of LAMP1 and ~88% of LAMP2 organelles colocalized with LysoT, suggesting no overall defects with lysosomal acidification. Additionally, we confirmed this finding using the SEP-LAMP1-RFP construct. We have further clarified this important issue in the revised manuscript (please see Figure 5—figure supplement 1 and accompanying text). To address the question of OPTN expression levels, as discussed above in our general response, we now provide images of non-transfected and Halo-OPTN overexpressed cells to illustrate representative levels of endogenous and overexpressed OPTN in the cells we chose for analysis (see Author response image 2 for images).

While we appreciate the reviewer’s comment that OPTN expression levels could accelerate the assembly of fully coated autophagosomes around damaged mitochondria, we argue that this is less relevant in our experimental conditions. In neurons, we see on average ~1-5 events per cell, which is low and distinct from the number of events during wholesale disruption of the mitochondrial network that is seen in HeLa cells treated with CCCP. As described in Figure 1, we are observing a minor fraction of the neuronal mitochondrial volume turnover. Additionally, we relied on endogenous levels of PINK1 and Parkin to initiate this pathway instead of overexpressed protein. If experimental conditions caused a substantial increase in the number of OPTN-coated mitochondria, then this point would become more relevant. We did look at cathepsin levels and observed no changes in our experimental paradigm. Importantly, we now include data to confirm that lysosomal acidification is normal (see Figure 5—figure supplement 1). Thus, it would appear that only mitophagolysosome acidification is slow. We plan to follow up on this interesting observation in the future.

With regards to the conclusions in Figure 5, an ultrastructure analysis in the cited Cheng et al., 2018 paper suggests that LAMP1 can be present on immature double membrane autophagosome-like structures. Is it possible that some of the LAMP1-positive signals lacking LysoT positivity are actually autophagosomes that are LAMP1 positive but not yet fused with a lysosome? Could LAMP1 overexpression affect the interpretation of the experiments?

Cheng et al., (2018) showed that there is a gradient in the number of LAMP1 organelles that contain Cathepsin D, with soma being higher than the axons and dendrites. In the Results for Figure 5, we argue that the observed discrepancy between LAMP1+/total and LysoT+/total may be in part due to the possibility that some of LAMP1 signal is on ‘late’ mitophagosomes (LC3LAMP1-positive) that have yet to fuse with degradative-competent organelles.

To address this issue more directly, we utilized a tandem SEP-LAMP1-RFP reporter where we could monitor acidified degradative-competent organelles. Using this method, we determined that the vast majority of OPTN rings around damaged mitochondria were dual labeled, indicating non-acidified compartments (see the new Figure 6).

Subsection “Low-level induction of ROS leads to selective mitochondrial damage without compromising the overall mitochondrial network”. In the two color old/new mitochondrial experiment, it's hard to imagine how the age can be precisely controlled using the described protocol. The SNAP tag for mito localization contains a COXVIII MTS sequence. As such, it would seem like any mitochondria with a functional translocon could import the SNAP-tag protein, regardless of whether it is old or new, however the authors state that the two populations are "distinct". The interpretation that old mitochondria can import the newly synthesized snap tag is consistent with most of the mitochondria being double labeled (Figure 6C). Because the mass of old mitochondria is much larger than the mass of new mitochondria that has been made since the new dye was added (dye added for 1 hour) and the number of those mitochondria not undergoing intermixing by fusion is small, the frequency with which such a "new" mitochondria would be identified as damaged and associated with OPTN would be particularly low, relative to the much more abundant old mito. So it seems possible, that this experiment doesn't really have the statistical power to rule out similar rates of old and new mitochondrial engulfment. The authors argue that the absence in green signal means that they are exclusively old, but it also may be that the signal to noise isn't equivalent. The labeling period for old was 24 hours while the labeling for new was only 1 hour. Therefore, all things being equal in terms of mitochondrial assembly/fission/fusion, there should be far less signal for new snap relative to old snap/per individual mito.

In Figure 7, the labeling period is the same for both ‘first’ and ‘second’ SNAP ligands (30 minutes of labeling, 2 quick washes, and a 30 minute washout). The first SNAP ligand labeled expressed Mito-SNAP from the time of transfection until the SNAP Block labeling, ~24 hours; this is termed the ‘old’ mito population. The second SNAP ligand labeled Mito-SNAP that was expressed from the time after SNAP Block until imaging, ~22 hours; this is termed the ‘young’ mito population. The dark SNAP Block was added to saturate any remaining binding sites after the first SNAP labeling (1 µM SNAP-block; 2 hours of labeling, 2 quick washes, and a 30 minute washout).

It is possible that mitochondria with a functional translocon could import the SNAP-tag protein and ligand, but we would argued that the damaged OPTN-positive mitochondria would not have the capacity to import new protein or be labeled by the new ligand because it is sequestered. Thus, these rounded structures would indicate that they were labeled and engulfed before the addition of the new SNAP-ligand. In the revised manuscript, we more clearly describe the labeling protocol in the text and figure legends and have updated the labeling schematic in the revised Figure 7.

In Figure 7, the authors examine an OPTN mutant found in ALS, and see general but small effects on mitochondrial size. This experiment is somewhat limited by the fact that OPTN deletion by siRNA is only ~60%, and it is possible that the overexpressed OPTN mutant (not sure how much it is overexpressed as they didn't show a western of the transfected cells) may also act as a dominant negative – binding TBK1 but not Ub chains. What isn't clear is whether the authors have missed an opportunity to statistically determine whether the OPTN mutant encircles damaged mito. The experiments are largely limited to mito morphology.

Please see the revised Figure 8—figure supplement 1C for a western blot showing relative equal levels of overexpression of Halo-OPTN WT and E478G. However, since neuronal transfection efficiency is low, it is not surprising that a 60% reduction in OPTN was observed. This reduction in OPTN expression is from an entire plate of cells, which includes cells that did not take up the siRNA. When imaging neurons, we selected cells that are transfected and therefore were more likely to also have taken up the siRNA.

Since it was previously shown that OPTN^E478G^ fails to engulf damaged mitochondria (Moore and Holzbaur, 2016), we sought to understand the functional significance of expression of this mutant on mitochondrial and neuronal health. Using our mitophagy-inducing paradigm, we think these experiments provide new insights into how expression of this mutant could lead to neurodegeneration.

An important weakness of the analysis is the absence of a direct demonstration of mitophagic flux.

We now include mitophagic (Cox8-EGFP-mCherry) flux assays and complementary autophagic (mCherry-GFP-LC3) and lysosomal (SEP-LAMP1-RFP) flux experiments, please see the new Figure 6, Figure 6—figure supplement 1, and Figure 5—figure supplement 1.

Reviewer #2:In their manuscript, Evans and Holzbaur present data arguing two main points regarding mitophagy in neurons: (1) that mitophagy occurs mainly in the soma and not in axons, contrary to some previous work by others, and (2) that neuronal mitophagy is significantly slower relative to other cell types, as result of slow acidification of mitoautophagosomes. Although the experiments are generally well-performed additional evidence is required in support of their main conclusions.Major comments:1) The implication throughout the paper is that the optineurin rings forming around mitochondria in response to mild oxidative stress are due to PINK1/Parkin dependent mitophagy. This should be demonstrated with knockdown or knockout of PINK1 and Parkin. This is particularly important given lack of clarity on the role of PINK1/Parkin mitophagy in neurons in response to more physiologically relevant stressors.

As previously stated, we performed Parkin knock-down and rescue experiments. We observed a 30% reduction in Parkin expression by immunoblot analysis. It is important to note that this measurement is an underestimate, as lysates were generated from an entire plate of hippocampal neurons, which transfect at a low efficiency and therefore includes many cells that did not take up the siRNA. When imaging neurons, we selected cells that were transfected and therefore did take up the Parkin siRNA. This level of knock-down was sufficient to significantly inhibit the number of cells with OPTN-positive rings, from ~35% down to less than 10%, which was rescued by re-expression of WT Parkin. This data has been included in the revised manuscript, please see Figure 2—figure supplement 2.

2) A major finding of the manuscript is that mitophagy occurs primarily in the cell soma and not the axons, in contrast to the findings of Ashrafi et al. As a rationale for this discrepancy, they suggest that they are applying a milder stress (withholding antioxidants from neuronal media) that does not lead to global bioenergetic collapse. This explanation doesn't seem satisfactory, however, as Ashrafi et al., also, used a variety of experimental setups to induce local damage mitochondria without affecting the overall bioenergetics of the neuron. An alternative explanation for the discrepancy might be that the particular exposure used by Evans and Holzbaur (withholding antioxidants) preferentially increases oxidative stress and mitochondrial damage in the cell soma as opposed to the axons. Does ROS increase equally in the axon and cell body with antioxidant withdrawal? Does the authors treatment result in TMRE negative mitochondria in axons that are not captured by Optineurin in contrast to the cell soma? When the authors treat with antimycin for 2 hours (which increases ROS locally at the mitochondria) do they see the same preferential increase in mitophagy in the soma or are mitochondria in the axons also targeted? Ashrafi saw axonal events as soon as 45 minutes, do the authors see axonal events at these early timepoints? Ashafri noted increased axonal events with the use of lysosomal inhibitors. If these are used by the authors, do they then see axonal events?

Ashrafi et al., (2014) induced mitochondrial damage using much higher doses of AA (20 or 40 µM) or locally via mito-KillerRed. In the mito-KillerRed experiments, imaging was initially performed in HBSS buffer instead of Hibernate E to enhance the effects of ROS produced by mito-KillerRed. We argue that this approach may affect the overall bioenergetics of the neuron.

We performed experiments to observe the ROS production of axonal mitochondria in neurons treated with antioxidant removal or by a much lower concentration of AA. We measured CellROX fluorescence intensity in the soma and processes of hippocampal neurons using fluorescence microscopy (note CellROX measurements in Figure 1B were performed on a plate reader). In these experiments, we observed significant increases in ROS production, measured by increases in CellROX intensity, in both the soma and processes of hippocampal neurons following 6 hours AO-free or 2 hours of 3 nM AA treatment (Figure 3G-H), suggesting that our global treatment targets all mitochondria. Further, since we observed shorter axonal mitochondria after 2 hours AA treatment, this timepoint should be sufficient to identify axonal events.

To follow up on this point, we measured axonal mitochondrial health using TMRE and saw no differences in TMRE intensity across these conditions (Figure 3—figure supplement 1J). However, this is not surprising as we did not observe differences in the somal mitochondrial population, where we routinely saw mitophagic events (Figure 1D). These observations are fully consistent with our approach of inducing low level damage without grossly perturbing the overall mitochondrial network.

We treated cells for 2 hours with AA and looked for colocalization of OPTN with fragmented mitochondria in the axon. Of the 21 neurons analyzed, we did not observe OPTN-positive mitochondria in the axon. Yet, similar treatments resulted in 35% of neurons having OPTN-positive mitochondria rings in the soma (Figure 3F), suggesting preferential increases in the somal compartment. Additional experiments were performed where AA treatments were reduced to 1 hour. Again, we observed mitophagic events were rare in the axon (3 events in 1 axon of the 28 neurons analyzed), but routinely found in soma (Figure 3D). However, we think it likely that if we did use a 10-20-fold higher dose of AA, we might see more evidence of axonal mitophagy as previously described (Ashrafi et al., 2014). It is possible that neurons are in a survival state, where axonal mitochondria undergo mitophagy locally to preserve the health of the neuron. We clarify this point in the revised manuscript. In general, however, under the paradigm described here axonal mitophagic events were rare across all conditions and did not increase with mild oxidative stress relative to control or with longer incubation times (1 vs 6 hours).

3) In Figure 5 the authors compare the speed of mitophagosome maturation in neurons and HeLa cells. They conclude that the difference in speed reflects a difference in processes downstream of mitochondrial ubiquitination, but the level mitochondrial ubiquitination is likely very different in neurons with endogenous Parkin subjected to mild oxidative stress and HeLa cells over-expressing Parkin and fully depolarized with OA. Could this account for the difference in maturation rate? Do the authors still see a difference in the speed of mitophagosome maturation when the two cell types are subjected to more equal treatment – e.g., in neurons and HeLa cells that are both overexpressing Parkin and are subjected to the same oxidative stress (e.g., antimycin)? Additionally, the authors note that acidification of mitoauthophagosomes is delayed in neurons? How long is it delayed? What do they see at time points longer than one day?

While we appreciate the point that we could use alternate paradigms to induce mitochondrial stress or attempt to equalize levels of Parkin expression to make HeLa cells a closer model of neuronal mitophagy, we thought that the more important question was to determine whether there is a universal deficit in autophagosome/lysosome acidification in hippocampal neurons subjected to mild oxidative stress, or whether there is a specific deficit in the acidification and eventual degradation of mitophagolysosomes. We therefore focused on this point. New data now included in the manuscript reveal that there is no overall perturbation in the extent of either autophagosomal or lysosomal acidification in neurons incubated in AO-free media as compared to control conditions (Figure 5—figure supplement 1). Thus, our data suggest that delayed acidification of mitophagosomes or ‘late’ mitophagosomes is specific in neurons. Using our experimental conditions, it would be difficult to continuously image cells for days. However, in Figure 7 we include data demonstrating the persistence of pulse-labeled OPTN-positive mitochondria that had yet to undergo autophagosome or lysosome engulfment 24 hours after mitophagy induction. Thus, the delay in acidification of mitophagolysosomes appears to be on the order of hours to days.

Reviewer #3:In the manuscript by Evans and Holzbaur, the authors examine the dynamics of OPTN recruitment to mitochondria and subsequent delivery to lysosomes following mild oxidative stress in primary rat hippocampal neurons. The authors mainly use transfected neurons and live-cell microscopy and find that OPTN is rapidly recruited to a subset of mitochondria, primarily in the soma, following anti-oxidant removal from the cell media. This then correlated with subsequent recruitment of mitophagy and lysosomal markers. Surprisingly, and the main take home message from the manuscript, was that the OPTN puncta on mitochondria do not become efficiently acidified (as visualized by col-localisation with lysotracker), from which the authors conclude that this is a rate-limiting step during neuronal mitophagy.This is an intriguing manuscript with many interesting observations. The identification of "mild" depolarising conditions that trigger a mitophagy response will be useful to the field and certainly does appear more physiological that the use of CCCP or oligomycin/antimycin. The microscopy images are impressive and convincing in what they show. However, I think more work is needed to determine whether this phenomenon is specific to mitophagosomes and neurons in general.Major comments:1) I think more robust analyses of lysosome acidification are needed. It is surprising and very interesting that there is a large delay in acidification of mitolysosomes. Is this a general phenomenon, i.e are acidified lysosomes scarce in these cells? What is the percentage of LAMP1 structures (regardless of co-localising with OPTN) that are LysoTracker +ve? If it is low, could this mean that most of the LAMP1-GFP is not in lysosomes? Could the authors use another lysosomal marker to confirm?

To investigate the acidification of lysosomes, we measured the percentage of LAMP1-EGFP structures that were positive for LysoT. We determined that ~89% of LAMP1 structures were also positive for LysoT (red arrows). In addition, we repeated these experiments with LAMP2GFP and observed ~88% of LAMP2 structures were also LysoT-positive. Thus, in our hands the majority of lysosomes in the soma are acidified in hippocampal neurons; representative images are shown in Author response image 4.

**Author response image 4. respfig4:** Representative images of hippocampal neurons illustrating that the majority of LAMP1-positive (**A**) or LAMP2-positive (**B**) organelles are acidified, as indicated by the presence of LysoT. LAMP-LysoT-positive lysosomes are indicated by red arrows and inset; LAMP-positive LysoT-negative organelles are indicated by open white arrows. Scale bar, 5 μm.

2) Related to the above point, the authors only use LysoTracker to estimate lysosomal acidification. Is there another way the authors could estimate this – for example using ratiometric fluorophores, such as Keima or mCherry-GFP?

As described above, we utilized SEP-LAMP1-RFP, a superecliptic pHluorin that is fused to the lumen domain of LAMP1-RFP (Farias et al., 2017) to measure lysosome acidification. In the soma, ~96% of LAMP1 organelles in control conditions were positive for RFP but negative for SEP, indicating that the majority of lysosomes are acidified organelles. Similar results were observed in 1 hour AO-free conditions. These data are now included in the revised manuscript, please see Figure 5—figure supplement 1.

3) Is the delay in acidification specific to mitolysosomes? What about autophagosomes in general? The authors could monitor the formation of OPTN -ve LC3 puncta under their conditions and determine if these equally take as long to become acidified. The authors may wish to discuss Ralph Nixon's work (e.g. Boland et al., 2008) that argues autophagosome fusion with lysosomes is very efficient in neurons.

We performed the requested experiment and determined that 71-85% of autophagosomes in the soma are acidified. Additionally, autophagosome acidification was not affected by antioxidant removal, with similar results found with either 1 or 6 h treatments (Figure 5—figure supplement 1). Furthermore, we observed a considerable amount of LC3-II via western blot across all conditions (Figure 1G-H). Collectively, these results suggest autophagosome fusion and acidification is efficient in neurons and support the findings of Boland et al., (2008).

4) Are the authors sure it is mitophagosome acidification that is rate-limiting and not an earlier step in the pathway? For example, is it possible that the autophagosome is not fully sealed (despite appearing circular by fluorescence microscopy). The authors could look at early autophagosome markers such as ULK1, ATG5 or WIPI that are not present on mature autophagosomes. Or mature autophagosome markers such as STX17.

To determine whether earlier steps in the pathway were rate-limiting, we observed WIPI2B, an early autophagosome marker, colocalization with mitophagosomes. In the soma, we observed WIPI2B puncta that colocalized with LC3, indicating these are immature autophagosomes. However, in both control and AO-free conditions OPTN-LC3-positive mitochondria were negative for WIPI2B, suggesting that these autophagosomes are fully formed.

**Author response image 5. respfig5:** Representative images of neurons 1 h after control or AO-free treatment. Mitophagosomes are indicated by red arrows and inset. OPTNLC3-positive mitochondria are negative for WIPI2B, indicating fully formed autophagosomes. Scale bar, 5 μm.

5) The authors imply that this delay is specific to neurons, yet they only look in hippocampal neurons and compare this to HeLa cells overexpressing Parkin. I think the authors need to provide some more data in relevant cell lines for a better comparison. What about cortical neurons, as well and comparing with primary fibroblasts (which should contain all the relevant Parkin pathway components)?

We thank the reviewer for their comment. The use of HeLa cells provides an alternative cell line to investigate mitophagy. Since HeLa cells are readily used to characterize mitophagy and we observed similar temporal dynamics for OPTN and LC3 translocation, we felt that it was a suitable system to compare for rates of mitophagosome acidification. Previous work has looked at the time course of mitophagy in cortical neurons where the authors suggested that mitophagy was slower and compartmentally restricted compared to non-neuronal cells (Cai et al., 2012), consistent with our findings – we further clarify this point in the revised manuscript.

[Editors' note: further revisions were requested prior to acceptance, as described below.]

The manuscript has been improved but there are some remaining issues that need to be addressed before acceptance, as outlined below:Based on the comments of reviewer 3, a couple of changes in the manuscript are still needed. First, some data that are described in the rebuttal don't appear in the paper, so Figure 4 may need to be updated to include data that were discussed in the rebuttal. In addition, there are a couple of suggested changes to the text that will need to be addressed.Reviewer #3:This is a re-review of the manuscript by Evans et al., describing the slow maturation/impaired acidification of mitophagosomes in the soma of neurons. I think the authors have done a great job in strengthening their manuscript – my main concern was with the lysosomal acidification studies and the use of the dual reporters means the data are much more convincing. I just have a few small points remaining:1) The LAMP2 data, mentioned in subsection” Acidification of OPTN-positive mitochondria is a rate-limiting step in neuronal mitophagy” and in the comments to reviewers does not seem to be in the manuscript. The authors mention Figure 4E, but these are only LC3-related. Please include the data.

In addition to providing the percent of LAMP1 and LAMP2 structures that were Lyso-T positive in the text, we now include images in the revised manuscript (please see Figure 5—figure supplement 1).

2) Likewise, the authors show data in the reviewers comments that the OPTN/LC3 structures are WIPI -ve. I do not see these data in the manuscript. This shows that the structures are not early autophagosomal in nature and thus they will be of interest to readers (and not just this reviewer) and so should be included (at least in the supplemental). I'm assuming WIPI2B is expressed in these cells and does indeed form puncta with LC3 when autophagy is induced?

At the reviewer’s request, we now include the WIPI2B data in Figure 5—figure supplement 1. WIPI2B, an early autophagosome marker, has been shown to be critical in the autophagy pathway, as depletion of WIPI2 in DRG neurons significantly decreased autophagosome biogenesis (Stavoe et al., 2019). In hippocampal neurons, we observed WIPI2B puncta that colocalized with LC3 in the soma, indicating immature autophagosomes (examples are highlighted by green arrows).

3) I also take on the point that the authors have made in response to my concern raised in point 5 – i.e. that HeLa cells overexpressing Parkin are a suitable system to compare to the neurons. The HeLa cell has been instrumental in the field in terms of building up the mechanism of PINK/Parkin mitophagy. But this is not a "normal" cell. Normally, it does not undergo this pathway as it does not express Parkin! I think to avoid doing more experiments, the authors can be more accurate in what they state and say that this delay in acidification does not occur in HeLa cells expressing Parkin. For example in subsection “Acidification of OPTN-positive mitochondria is a rate-limiting step in neuronal mitophagy”: "…..much slower than the corresponding time course in non-neuronal cells." Should be replaced with "…..much slower than the corresponding time course in HeLa cells."

The statement has been changed to “much slower than the corresponding time course in HeLa cells expressing exogenous Parkin.”